# The psychology of professional and student actors: Creativity, personality, and motivation

Denis Dumas[1]*, Michael Doherty[2], Peter Organisciak[1]

**1** Department of Research Methods and Information Science, University of Denver, Denver, Colorado, United States of America, **2** Actor's Equity Association, New York, NY, United States of America

* Denis.Dumas@du.edu

## Abstract

As a profession, acting is marked by a high-level of economic and social riskiness concomitantly with the possibility for artistic satisfaction and/or public admiration. Current understanding of the psychological attributes that distinguish professional actors is incomplete. Here, we compare samples of professional actors ($n = 104$), undergraduate student actors ($n = 100$), and non-acting adults ($n = 92$) on 26 psychological dimensions and use machine-learning methods to classify participants based on these attributes. Nearly all of the attributes measured here displayed significant univariate mean differences across the three groups, with the strongest effect sizes being on Creative Activities, Openness, and Extraversion. A cross-validated Least Absolute Shrinkage and Selection Operator (LASSO) classification model was capable of identifying actors (either professional or student) from non-actors with a 92% accuracy and was able to sort professional from student actors with a 96% accuracy when age was included in the model, and a 68% accuracy with only psychological attributes included. In these LASSO models, actors in general were distinguished by high levels of Openness, Assertiveness, and Elaboration, but professional actors were specifically marked by high levels of Originality, Volatility, and Literary Activities.

## Introduction

In much of the industrialized world, where access to entertainment has become nearly ubiquitous for many individuals, professional actors constitute a rarified population of experts who receive high levels of attention in the popular press and in public discourse. The work of a small number of professional actors reaches a relatively large swathe of the public, and the average person may be exposed to the work of a professional actor far more often than they are to the efforts of most other kinds of experts (e.g., medical or legal professionals). However, despite the fact that it is possible for actors to reach high-levels of wealth and notoriety, the vast majority of professional actors—even those who work consistently—make a modest wage and have a paycheck-to-paycheck lifestyle more akin to blue-collared workers, rather than other professionals [1]. In addition, most actors experience economic uncertainty throughout their careers, with large temporal gaps in their employment, which can occur regardless of their past success [2]. In addition, the prospect of achieving professional success for

the text-mining based models used to score the divergent thinking tasks are freely available on our laboratory website (https://openscoring.du.edu/).

**Funding:** The author(s) received no specific funding for this work. Actor's Equity Association did not provide any financial support for this study, including in the form of author salary. As a collective authorship team, we have no competing interests that could interfere with the scientific process related to this work.

**Competing interests:** Michael Doherty's affiliation with Actor's Equity Association did not and does not alter our adherence to PLOS ONE policies on sharing data and material.

undergraduate-level student actors is also far worse than that of most other undergraduate majors (e.g., biology, engineering), with the majority of acting majors leaving the profession within a few years of college graduation [3].

Given this current characterization of the acting profession, with its extremely high-risk features, the question becomes pertinent: what psychological attributes distinguish those individuals who have dedicated themselves to the acting profession from those that have not? And when professional and student actors are jointly considered, what psychological attributes may differentiate those groups, and potentially contribute to professionals' persistence and success? By closely examining and working to understand the psychological attributes that support actors in their professional work, or their development of expertise as a student, it is our intention to present findings that are not only interesting for the psychological research community but also for professional and student actors themselves. In addition, educators who seek to train student actors who may one day become professional may benefit from such an investigation, because it intends to highlight the dimensions on which professional and student actors differ or are similar, possibly informing pedagogical decisions. Finally, organizations that employ, serve, or represent actors (e.g., theater companies; talent agencies; labor unions) may find value in such an investigation, because it would delineate the strengths and further needs of actors and the acting community.

In order to best accomplish these general aims, we here analyze psychometric data from three groups of individuals: non-acting adults, undergraduate-level student actors, and professional actors who have already achieved a recognized degree of success. From each of these three groups, creative, motivational, and personality attributes are measured, and machine learning methods are utilized to identify psychologically-relevant patterns in these data. But before presenting the current study, we first overview the existing knowledge concerning the psychology of professional actors and highlight areas in which these existing studies can be augmented to provide a richer, and more modern, understanding.

## Summary of extant empirical psychological work with professional actors

Actors and acting have been of interest to psychologists since the nascent beginnings of the field, with major early psychologists such as Binet [4] and Vygotsky [5], writing theoretical pieces on the psychology of the performing arts (Binet even co-wrote and produced several popular French plays [6]). Despite this historical interest among psychologists in actors and acting theoretically, the actual body of empirical work dedicated to actors is small compared to investigations into other areas of expertise (e.g., engineering; [7]). The earliest empirical investigation of the psychology of professional actors of which we are aware was conducted by Stacey and Goldberg [8]), who used a number of self-report psychometric scales to show that undergraduate student actors who were regularly cast in university productions were more psychologically similar to professional actors than were undergraduate student actors who were rarely cast in university productions. Specifically, Stacey and Goldberg [8] showed that student actors who were rarely cast exhibited much higher levels of extraversion than did students who were regularly cast or professional actors, who were much more inhibited, conscientious, and prone to depressive thinking.

This early finding paved the way for the continuing investigation of negative psychological attributes in performing artists: a program of research that has been particularly productive in recent years. For example, Dufner and colleagues [9] found that, among dedicated (but not professional) improvisational actors, heightened levels of a narcissistic need for admiration can be observed. However, the improvisational actors showed much lower levels of a narcissistic need to derogate others (known as rivalry), demonstrating that actors may be

narcissistically motivated towards self-promotion, but that tendency did not extend to putting other actors down. In related qualitative work, Robb, Due, and Venning [10] conducted in-depth interviews of self-identified professional actors concerning their well-being and vulnerability to mental illness (e.g., anxiety and depressive disorders), finding that professional actors used a wide-range of strategies to protect their well-being including positive engagement with the artistic community, focusing on personal growth, and conceptualizing acting as a meaningful life-purpose. Davison and Furnham [11] used a large sample of professional actors with strict inclusion criteria (i.e., actors were recruited through their agents, assuring they were indeed professional) and administered a number of well-validated self-report measures that were aligned with personality disorder profiles (e.g., Schizoid, Dependent, Obsessive-Compulsive). In general, they identified heighted subclinical levels of personality disorder related traits in their sample: with actors scoring significantly higher than non-actors on Antisocial, Narcissistic, Histrionic, Borderline, and Obsessive-Compulsive personality disorder scales, and only male actors displaying significantly heightened levels of Schizotypal, Avoidant, and Dependent personality disorders. This finding appears to be related to Thomson and Jaque's [11] finding that professional performing artists (including actors and dancers) who reported significantly greater amounts of Adverse Childhood Experiences (ACEs) also reported experiencing creative states (e.g., a transformational sense of self) more often than performing artists who did not experience as many ACEs.

Using a relatively large sample of self-identified professional actors (82% were members of an actor's union), Nettle [12] administered questionnaires designed to tap the Big 5 personality dimensions (i.e., Neuroticism, Extraversion, Openness, Conscientiousness, Agreeableness [13]) as well as psychological attributes related to autism spectrum disorder (i.e., Empathizing Quotient and Systemizing Quotient; [14]). Nettle [12] compared actors' scores on these measures to general population British norms and found that actors scored significantly higher than the normative level on Extraversion, Openness to Experience, and Agreeableness, and had heightened but non-significantly different levels of Neuroticism. The actors also scored significantly higher than the comparison norms on the Empathizing Quotient but were similar to the British norm on the Systemizing Quotient. This general picture of the psychology of professional actors has also led to the use of actors as participants in the investigation of a number of other psychological processes and conditions such as facial recognition of emotions [15] post-traumatic stress disorder [16], and neurological effects of auditory-motor expertise [17]. In addition, some psychological phenomena that specifically affect actors—such as stage fright—have been examined using professional actors as participants [18]. Specifically, these researchers found that, among professional actors, females with low emotional stability and an external locus of control were most at risk for serious and recurring stage fright.

Another recent study specifically of student actors [19] examined the emotional attributes of undergraduate acting majors as compared to undergraduate students without acting experience. These researchers found that actors reported higher temperamental sadness and fear, but more positive viewpoints related to the experience of these negative emotions. In addition, student actors were more capable than other undergraduates at identifying facial expressions related to pride, but less capable than other undergraduates at identifying facial expression related to anger. This finding is related to another recent piece from Ivcevic and colleagues [20] who found that, despite the strong negative correlation at the population level between psychological vulnerabilities such as anxiety and depression and psychological resources such as self-acceptance and hope, creative experts (i.e., fine arts faculty) exhibited simultaneously high levels of both psychological vulnerabilities and resources, implying that creative experts may be fruitfully utilizing both their negative *and* positive psychological attributes to support their artistic expression. These findings are supported by a relatively long line of psychological

research from scholars such as Thalia Goldstein and Ellen Winner [21–24] who have shown that arts education, and specifically training in acting techniques, can support children's development of emotional regulation, theory of mind, and other positive psychological attributes, including the capacity to safely express negative emotions. In our view, these perennial findings from the developmental and educational literature concerning the benefits of acting training for children further imply that expert of professional actors may not only benefit from such positive psychological attributes that they develop during their training but may actually require those attributes for success in their expert work.

These inferences regarding the expertise of actors are highly related to a line of acting research situated within the literature on expertise development. Noice & Noice [25–27] conducted a series of studies on the cognitive processes that actors use when preparing for their roles. This line of research eventually culminated in Noice & Noice [28, 29] positing a two-stage model of actors' process: *analysis* and *active experiencing*. For Noice and Noice, each of these major stages of the acting process were further broken down into a number of component processes that were often reminiscent of the information-processing perspective on expertise development [30]. For example, in Noice and Noice's model, finely-grained sub-processes such as causal attribution (when actors define 'why' something occurs in a script) can occur many times within the analysis phase of role preparation, depending on the demands of the project and expertise of the actor. In a related but much more recent line of work [31], psychologists have also begun to examine the specific vocal strategies that actors use to embody characters with differing personality traits (e.g., Assertiveness, Cooperativeness), finding that actors altered their voices along 12 vocal parameters (e.g., pitch, volume) in order to portray the personality of their characters.

## Promising areas to extend past work

Given the extant understanding of the psychology of professional actors that is present in the field, a number of opportunities to extend, improve, and also replicate existing work are apparent. Here, we briefly delineate areas in which the current state-of-the-art in research on professional actors can be moved forward.

**Divergent thinking assessment.** One clear pattern that is discernible in the current literature is that there has been a heavy reliance on self-report measures in research on professional actors. This choice is understandable given time and resource limitations in this area of work, but the greater inclusion of performance measures in this line of research remains a strong opportunity. In the larger literature on creativity and creative expertise, the most commonly administered performance measures are Divergent Thinking (DT) assessments, which require participants to generate multiple possible solutions to a given task within a set amount of time [32]. These tasks are typically scored along multiple dimensions that generally correspond to the quantity (i.e. Ideational Fluency) and quality (i.e. Originality) of the ideas generated by a participant, with each of these dimensions having been repeatedly demonstrated to be strong positive predictors of creative potential and performance (see [33] for a review of DT scoring). Given the obvious demand of the acting profession to generate interesting or original ideas rapidly (such as when rehearsing a new role), DT can readily be hypothesized as relevant to the success of professional actors. However, the predictive power of DT to identify those individuals who are or could be professional actors remains unknown in the field. As far as we are aware, DT assessments have not yet been systematically administered to professional actors as part of research study, however, some initial evidence that DT measures are sensitive to acting training is available in the field. For example, Sowden and colleagues [34] demonstrated that improvisation exercises could improve the DT of elementary school students, suggesting that

DT measures may be suitable for identifying individuals with acting training. In this investigation, performance assessment of DT is included, as a key way to extend past work.

**Richer array of self-report questionnaires.** Although many psychologically interesting and well-validated self-report scales have been previously included in research on actors, many relevant constructs remain to be included. For example, self-reported creative activities in domains in which an individual is not a professional (e.g., creative visual arts activities for actors) have previously been shown to be predictive in creativity research [35]. For instance, it may be reasonable to hypothesize that actors, given the creative nature of their work, will engage in more creative activities than non-actors even in domains (e.g., literature; music) that are not directly within the area of acting. Relatedly, it could also be, that because actors' work demands creative thinking, they may tend to avoid expending creative effort in other more quotidian domains such as cooking.

In addition, motivational constructs such as Grit [36] that have come to greater attention in the literature recently, have never been examined with professional actors before. In the context of the acting profession, where financial security can be lacking, and rejection (i.e., not booking an auditioned-for role) is commonplace, motivational attributes such as perseverance in the face of adversity and consistency of interest in one's chosen profession—two principal facets of Grit [36]—appear likely to be relevant. Emotional Intelligence [37] also appears to be a candidate for relevance to professional actors, in that actors' work regularly consists of appraising and using emotions. Finally, although the Big 5 personality attributes have previously been investigated in professional actors [12], the more finely-grained analysis of personality facets (two of which load on each of the Big 5 [38]) has never before been examined with actors. More specifically, the Big 5 dimensions of Neuroticism, Agreeableness, Conscientiousness, Extraversion, and Openness can be further delineated into 10 facets: Neuroticism contains both Volatility and Withdrawal; Agreeableness is divided into both Compassion and Politeness; Conscientiousness contains Industriousness and Orderliness; Extraversion has Enthusiasm and Assertiveness; and Openness is divided into Intellect and Openness. In this study, each of these extensions to past work are included.

**Inclusion of student and professional actors.** Many scholars who have studied actors and acting have been interested with the process by which student or trainee actors become professionals and subsequently develop expertise [29, 31]. However, only rarely in the literature have student actors and more expert professional actors been specifically compared on their psychological attributes. In addition, within the general literature on creativity, psychological differences among individuals who are professionally creative and those who are not has perennially been of interest [39]. Regarding the specific question of how student and professional actors differ psychologically, the classic study—now nearly 70 years old—by Stacey and Goldberg [8] remains potentially the most informative. In the current study, we adopt a sampling strategy similar to Stacey and Goldberg [8] in that we specifically compare student and professional actors but modernize the work through the large-scale data collection that is now the standard of the field [40].

**Modern machine learning methodology.** Some existing studies of professional actors (e.g., [12, 40]) have collected impressively large samples given the challenges in recruiting this population for psychological research. However, within this literature, sample sizes have not necessarily been fully leveraged through the application of cutting-edge methodologies. Indeed, nearly all quantitative work in the study of professional actors have solely utilized traditional mean-comparison methods (e.g., ANOVA) or ordinary-least-square predictive methods (e.g., regression). Although these perennially applied methods are not inherently flawed, the opportunity to methodologically modernize the methods used to investigate the psychology of professional actors seems apparent. Specifically, with more psychological research

incorporating advanced methodologies from the machine-learning family [41] opportunities to potentially illuminate more nuanced, more generalizable, and possibly more replicable (i.e., not over-fit to a single dataset) patterns in our data become apparent. In this study, we used machine-learning methodology in two different ways. First, the DT assessment is scored using a modern computational scoring method derived from the text-mining literature [42], and Least Absolute Shrinkage and Selection Operators (LASSOs [43]) are utilized to provide the most psychologically informative prediction and classification models.

### Research questions of current study

Given the current state-of-the-literature reviewed here concerning empirical psychological research with professional actors, as well as the previously identified areas to extend that extant work, the current study posits three specific research questions to be investigated:

1. How do non-actors, student actors, and professional actors differ on average on a number of theoretically-relevant psychological attributes?

2. Can actors be effectively distinguished from non-actors based on their psychological attributes? What psychological attributes would be strongly-weighted by a model used for this purpose?

3. Can professional actors be effectively distinguished from undergraduate student actors based on their psychological attributes? What psychological attributes would be strongly-weighted by a model used for this purpose?

## Methodology

To facilitate replicability and open science, the data collected and analyzed in this study is archived at Zenodo (LINK: https://zenodo.org/record/3899579#.X3-M81KSlaQ; DOI: 10.5281/zenodo.3899578), and the text-mining based models used to score the divergent thinking tasks are freely available on our laboratory website (https://openscoring.du.edu/).

### Participants

A total of 296 individuals participated in this study, with participants being drawn from three different groups: (a) non-acting adults, (b) undergraduate students majoring in acting, and (c) adult professional actors. Each of these groups of participants are described separately in this section. All participant recruitment strategies used here, as well as study procedures, were approved by the Institutional Review Board at the University of Denver.

**Non-acting adults.** This group included 92 (53 female; 57.6%) participants. Non-acting adults were recruited for this study via Amazon Mechanical Turk, a crowdsourcing platform widely used in psychology research, including creativity research [44]. Because of the high language demands of divergent thinking tasks, participants were required to report themselves as fluent English speakers in order to participate, although 2 participants (2.1%) reported English as their second (but fluent) language. Participants were compensated $3.00 each for their participation. Participants were required to be over the age of 18 to participate, but the minimum actual participant age was 21, with a maximum age of 68. The mean age of participants was 37 ($SD$ = 10.58). The majority of participants ($n$ = 68; 73.91%) reported their race/ethnicity as White or European-American, while smaller proportions of the sample reported their ethnicity as Black or African-American ($n$ = 6; 6.5%), Asian ($n$ = 9; 9.8%), Latinx ($n$ = 5; 5.43) or multiple ethnicities ($n$ = 4; 4.2%).

Although this sample was collected as a non-acting comparison group, we did not require that these participants should have had zero history of activities within the performing arts. Indeed, some history of "little-c" [45] creative activities is likely to be expected of nearly any sample. However, as will be presented in the Results section of this paper (see Table 2 for standardized descriptive statistics), this comparison group did report statistically and practically significantly fewer creative activities than the other groups for every creative domain measured, with the greatest differences being within the performing arts domain.

**Undergraduate acting majors.**   100 undergraduate students, currently enrolled as acting or theater majors, participated in this study. Recruitment for this group of participants was primarily accomplished via existing social media listservs that connected undergraduate theater and acting students, although snowball sampling methods, in which participating students shared the participation opportunity with their classmates, were also utilized. The average age of the undergraduate students was 20.33 ($SD$ = 2.65), with a minimum age of 18 and a maximum age of 26. The sample was relatively evenly split among students in terms of the years they had spent in their undergraduate program: of the 87 student actors who reported their year-in-program, 24 (27.59%) were in their first year, 15 (17.24%) were in their second year, 18 (20.69%) were in their third year, 19 (21.84%) were in their fourth year, and 11 (12.64) were in their fifth year as undergraduates. In addition, although all undergraduate actors had received targeted acting training as part of their education, they were situated in three different concentrations within their programs: of the 78 student actors who reported their concentrations 35 (44.87) were in an acting concentration, 27 (34.61%) were in musical theater, and 16 (20.51%) were in a directing, playwriting, or production concentration. 77 (77.00%) students reported a female gender, 18 (18.00%) reported male, and 5 (5.00%) reported a non-binary gender identity. 77 (77.00%) reported an ethnicity of White or European-American, 6 (6.00%) reported their ethnicity as Black or African-American, 2 (2.00%) Asian, 4 (4.00%) Latinx, and 11 (11.00%) multiple ethnicities. 95% ($n$ = 95) reported being a native English speaker, with 5% ($n$ = 5) reporting a first language other than English.

**Professional actors.**   In this study, 104 professional actors were recruited via existing email and social media listservs that connected members of professional actor's unions: either Actors Equity (which focuses on the representation of stage actors) or SAG-AFTRA (which focuses on the representation of screen actors). During the ongoing data collection process, some snowball sampling of professional actors occurred, where individual actors would share the participation opportunity with their colleagues. In order to be eligible to participate, actors needed to report being a member of either Actor's Equity of SAG-AFTRA, or if they were not union-affiliated, report having previously booked more than 10 professional acting contracts either on stage or screen. In addition, 56.38% ($n$ = 53) reported holding a Bachelor's degree in acting, while 29.79% ($n$ = 28) reported holding a Master's degree, and 3.19% ($n$ = 3) reported holding a doctoral degree. 5 participants (5.32%) reported having no university training, and the same proportion reported having a non-degree certification from a university. 72.34% ($n$ = 68) also reported having engaged in additional acting training at a studio apart from their university training.

In terms of gender, 51.92% ($n$ = 54) of the professional actors reported female, 45.19% ($n$ = 47) reported male, .96% ($n$ = 1) reported a non-binary gender, and 1.92% ($n$ = 2) preferred not to respond. The average age of the sample was 35.43 ($SD$ = 10.13) with minimum age of 21 and a maximum age of 67. The majority of participants ($n$ = 84; 70.77%) reported their race/ethnicity as White or European-American, while smaller proportions of the sample reported their ethnicity as Black or African-American ($n$ = 3; 2.88%), Asian ($n$ = 1; .96%), Latinx ($n$ = 6; 5.77%) or multiple ethnicities ($n$ = 6; 5.77%). 97.12% ($n$ = 101) of the

professional actors reported they were native English speakers, with 2.88% (*n* = 3) reporting a first language other than English.

## Measures

In order to capture a variety of meaningful psychological attributes that may explain individuals' continued motivation and ability to engage with the acting profession, a number of well-validated measures were administered to participants. All scales administered here were scored using confirmatory factor analysis (CFA) models and empirical Bayes, in order to most validly quantify the psychological attribute tapped by the measure [46]. As such, three reliability indices are available in Table 1 for each of the measures: Cronbach's alpha [47], McDonald's Omega [48], and Hancock's *H* [49]. These reliability indices differ in that alpha assumes a tau-equivalent model (i.e. all item loadings equal) and therefore represents a lower-bound of the scale reliability [50]. Omega allows for assumption of tau-equivalence to be relaxed (i.e., item loadings can differ), and also accounts for the size of the error terms in a factor model [51] *H* is also referred to as *maximal reliability* [52], because it provides an upper-bound on the reliability of the scale and is most appropriate when measure scores are saved directly from a latent measurement model (as is the case in this study). Details about the measures appear below.

**Alternate Uses Task (AUT).**   The AUT is a psychometric measure in which participants are asked to generate as many creative uses for an object as possible within a certain amount of time (i.e., two minutes per object in this case). The AUT has been used for assessing divergent thinking and creative potential for decades [53–55], and remains one of the most-often utilized tasks within the creativity research literature [56, 57]. The following 10 object names were presented to participants in a randomized order: *book*, *fork*, *table*, *hammer*, *pants*, *bottle*, *brick*, *tire*, *shovel*, and *shoe*. The resulting AUT data are both open-ended (i.e., participants can differ on how many responses they give within the time limit) and ill-defined (i.e., participants can also differ on the number of words used to describe their responses), and therefore the AUT has typically been scored using a number of different scoring procedures, each designed to estimate a different dimension of divergent thinking [58, 59]. In this study, three scoring procedures were utilized for the AUT, and each are described below.

*Ideational Fluency*. Ideational Fluency refers to an individual's capacity to rapidly generate a number of ideas within a set amount of time [58]. Therefore, the scoring procedure for Fluency is relatively simple: the number of responses generated by each participant for each AUT item was tallied. These ten item-level Ideational Fluency scores exhibited a high level of composite reliability (see Table 1 for alpha), and when a unidimensional confirmatory factor analysis (CFA) model was fit to those item-level Fluency counts, they also showed a high level of factor reliability (see Table 1 for Omega and *H*). Fluency scores for each participant in the dataset were estimated via the CFA model using empirical Bayes, and saved for later analysis.

*Elaboration*. Elaboration refers to the degree that participants explicate their responses to the AUT [60] and within the creativity research literature Elaboration is commonly scored using word counts [61]. Here, the number of words participants utilized within each AUT item was counted, producing ten item-level Elaboration scores. These scores displayed a strong level of reliability both at the composite and latent factor levels and participants' Elaboration scores for the AUT were generated from a unidimensional CFA model using empirical Bayes.

*Originality*. Typically regarded as the most theoretically important dimension of divergent thinking measured by the AUT, Originality refers to the relative unusualness or novelty of participant responses [62]. A number of different scoring procedures for Originality exist in the creativity research literature, and this study follows the most modern methodological guidelines available [33]. Specifically, Originality is scored here via a text-mining approach, using

**Table 1. Reliability indices for measures used in this investigation.**

| Measure / Dimension | Reliability | | |
|---|---|---|---|
| | α | ω | *H* |
| Alternate Uses Task | | | |
| Fluency | .962 | .964 | .965 |
| Elaboration | .966 | .967 | .968 |
| Originality (mean) | .858 | .874 | .903 |
| Originality (max) | .824 | .834 | .842 |
| Inventory of Creative Activities | | | |
| Literature | .797 | .804 | .821 |
| Music | .876 | .881 | .945 |
| Crafts | .873 | .882 | .953 |
| Cooking | .872 | .879 | .899 |
| Visual Art | .817 | .826 | .847 |
| Performing Arts | .831 | .836 | .902 |
| Short Grit Scale | | | |
| Grit | .807 | .811 | .863 |
| Intolerance of Uncertainty Scale | | | |
| Prospective Uncertainty | .866 | .868 | .871 |
| Inhibitory Uncertainty | .877 | .878 | .887 |
| Emotional Intelligence Scale | | | |
| Self-emotion Appraisal | .886 | .894 | .909 |
| Other-emotion Appraisal | .913 | .917 | .940 |
| Uses of Emotion | .864 | .874 | .903 |
| Big 5 Aspects Scale | | | |
| Openness | .876 | .882 | .893 |
| Intellect | .846 | .855 | .892 |
| Enthusiasm | .924 | .927 | .933 |
| Assertiveness | .931 | .935 | .946 |
| Industriousness | .879 | .886 | .891 |
| Orderliness | .889 | .894 | .907 |
| Compassion | .952 | .953 | .957 |
| Politeness | .827 | .832 | .852 |
| Volatility | .941 | .942 | .945 |
| Withdrawal | .920 | .921 | .924 |

the *Global Vectors for Word Representation* (GloVe) *840B* system, which is publicly available through the Stanford natural-language-processing laboratory [63]. This text-mining model was trained on a corpus of 840 billion words that were scraped from a variety of online sources including Wikipedia and Twitter. Previous psychometric work in creativity research [42] showed that, among a number of candidate text-mining models, GloVe was the most capable of approximating human-rated Originality, displayed the most advantageous reliability coefficients, and had the most theoretically meaningful correlations to relevant criteria measures. In addition, in accordance with current methodological recommendations within creativity research [61] an inverse-document-frequency (IDF) weighting scheme was applied to the GloVe scoring system, in which the words utilized in participant responses were weighted more strongly if they were rare in model's training corpus and weighted more weakly if they were common in the model's training corpus [64].

**Table 2. Means, standard deviations, and univariate mean comparisons.**

| Variable | Non-Acting Adults | Undergraduate Acting Majors | Professional Actors | One-way Mean Comparison |
|---|---|---|---|---|
| **Fluency** | -.38 (.78) | .07 (.94) | .27 (1.11) | $F(2,293) = 11.99$, $p < .001$, $\eta^2 = .08$ |
| **Elaboration** | -.28 (.77) | .05 (1.02) | .20 (1.09) | $F(2,293) = 5.94$, $p = .003$, $\eta^2 = .03$ |
| **Originality (mean)** | .06 (.93) | -.29 (1.22) | .23 (.71) | $F(2,293) = 7.39$, $p < .001$, $\eta^2 = .05$ |
| **Originality (max)** | -.19 (.95) | -.04 (1.02) | .21 (.98) | $F(2,293) = 4.33$, $p = .014$, $\eta^2 = .03$ |
| **Literary Activities** | -0.81 (.91) | .23 (.81) | .50 (.78) | $F(2,293) = 66.41$, $p < .001$, $\eta^2 = .31$ |
| **Musical Activities** | -.86 (.79) | .38 (.81) | .39 (.83) | $F(2,293) = 74.50$, $p < .001$, $\eta^2 = .33$ |
| **Crafting Activities** | -.69 (.99) | .32 (.85) | .30 (.83) | $F(2,293) = 40.96$, $p < .001$, $\eta^2 = .22$ |
| **Cooking Activities** | -.33 (1.00) | .35 (.94) | -.05 (.95) | $F(2,293) = 12.64$, $p < .001$, $\eta^2 = .08$ |
| **Visual Art Activities** | -.57 (.91) | .31 (.85) | .20 (.99) | $F(2,293) = 25.59$, $p < .001$, $\eta^2 = .22$ |
| **Performing Arts Activities** | -.92 (.61) | .42 (.85) | .40 (.85) | $F(2,293) = 91.45$, $p < .001$, $\eta^2 = .38$ |
| **Grit** | -.32 (1.15) | .04 (.92) | .25 (.84) | $F(2,293) = 7.91$, $p < .001$, $\eta^2 = .05$ |
| **Intolerance of Uncertainty (Prospective)** | .40 (1.01) | -.35 (.90) | -.02 (.95) | $F(2,293) = 14.70$, $p < .001$, $\eta^2 = .09$ |
| **Intolerance of Uncertainty (Inhibitory)** | .24 (1.09) | -.31 (.88) | .09 (.94) | $F(2,293) = 8.60$, $p < .001$, $\eta^2 = .05$ |
| **Self-emotional Appraisal** | .02 (.97) | .18 (.86) | -.19 (1.11) | $F(2,293) = 3.71$, $p = .03$, $\eta^2 = .02$ |
| **Other-emotional Appraisal** | -.39 (1.14) | .32 (.69) | .03 (.99) | $F(2,293) = 13.54$, $p < .001$, $\eta^2 = .08$ |
| Uses of Emotion | .11 (1.02) | .08 (.89) | -.17 (1.05) | $F(2,293) = 2.55$, $p = .08$, $\eta^2 = .02$ |
| **Openness** | -.73 (1.20) | .32 (.71) | .34 (.62) | $F(2,293) = 35.95$, $p < .001$, $\eta^2 = .24$ |
| **Intellect** | -.23 (1.12) | .22 (1.02) | -.03 (.81) | $F(2,293) = 4.99$, $p = .007$, $\eta^2 = .03$ |
| **Enthusiasm** | -.47 (1.10) | .24 (.85) | .19 (.89) | $F(2,293) = 17.11$, $p < .001$, $\eta^2 = .10$ |
| **Assertiveness** | -.59 (1.21) | .31 (.71) | .23 (.78) | $F(2,293) = 28.17$, $p < .001$, $\eta^2 = .16$ |
| **Industriousness** | .42 (1.06) | -.17 (.85) | -.21 (.96) | $F(2,293) = 13.19$, $p < .001$, $\eta^2 = .08$ |
| Orderliness | .10 (1.21) | -.14 (.84) | .05 (.92) | $F(2,293) = 1.67$, $p = .191$, $\eta^2 = .01$ |
| **Compassion** | -.42 (1.35) | .29 (.67) | .09 (.75) | $F(2,293) = 14.37$, $p < .001$, $\eta^2 = .08$ |
| Politeness | .07 (1.15) | -.06 (.90) | -.02 (.94) | $F(2,293) = 0.52$, $p = .593$, $\eta^2 < .01$ |
| **Volatility** | -.24 (1.21) | -.03 (.91) | .23 (.84) | $F(2,293) = 5.57$, $p = .003$, $\eta^2 = .04$ |
| **Withdrawal** | -.24 (1.20) | -.02 (.88) | .22 (.86) | $F(2,293) = 5.72$, $p = .004$, $\eta^2 = .04$ |

*Note*: Groups means are presented in cells, SD's are in parentheses. All variables measured here are standardized across the entire sample: the grand mean is zero and grand SD is 1. But, means and SD's can differ across the groups, and therefore the comparisons here are relevant. Bolded variable names indicate that variable exhibited significant differences across groups at the $p < .05$ level.

Conceptually, GloVe is designed to preserve the linearity of the relations among words, so that the semantic distances between them are directly comparable by studying the factorization matrix of words by latent dimensions. Using these matrices, the latent dimensions can be used as coordinates in a geometrically represented space, and the cosine of the angle between the word-vectors can be interpreted as the semantic or associative distance among words [65]. As an example with the AUT, if the prompt was "shovel", the response "dig a hole" would result in a vector that has an acute angle with the vector for "shovel". In contrast, the response "fling tennis balls for a dog to chase" would result in a vector that has a wider angle from the initial prompt vector, indicating that response is less semantically similar. Please see Fig 1 for a visualization of the specific geometric relations among these example responses. The cosines of these angles among word vectors yielded semantic similarity indices that ranges from -1 to 1 for each AUT response, which were subtracted from 1 in order to yield Originality scores that ranged from 0 to 2. Originality scores were then averaged across the responses generated for each AUT item (e.g., Book), resulting in ten item-level Originality scores for each participant. These scores displayed a satisfactory level of reliability at both the composite-scale and latent

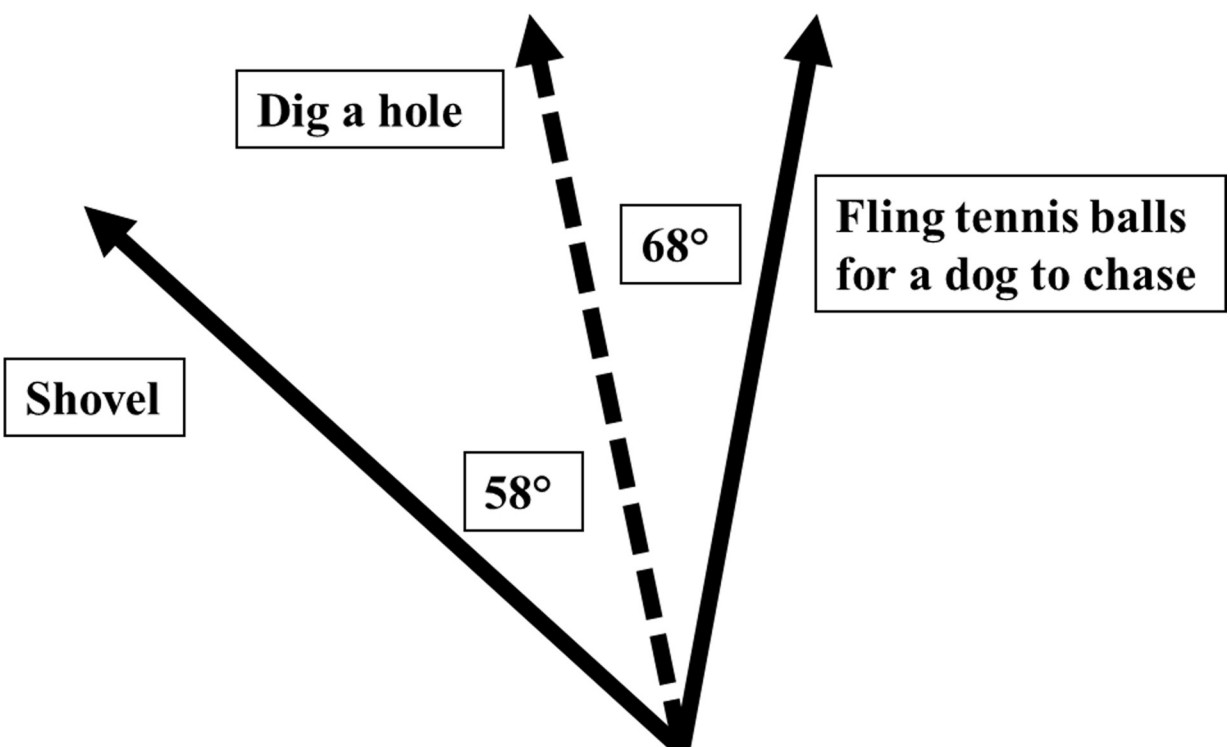

**Fig 1. A visual representation of the geometric relations among response vectors arising from a _GloVe 840B_ scoring analysis.** As can be seen, the more original a response, the greater its angle with the AUT prompt.

factor levels and Originality scores were saved for each participant from the unidimensional CFA model using empirical Bayes.

**Inventory of creative activities.** This inventory is a relatively recently developed [35] self-report measure for real-life creative activities and accomplishments across eight domains: literature, music, arts and crafts, cooking, sports, visual arts, performing arts, and science and engineering. Given the general nature of this sample, and time-constraints on the data collection, we administered the scales for six of those original eight domains: music, literature, arts and crafts, cooking, visual arts, and performing arts. Each of these scales consisted of six Likert-style items that ask participants how many times they have done particular creative activities in the past 10 years with five response categories: _never_, _1–2 times_, _3–5 times_, _6–10 times_, and _more than 10 times_. For example, in the music domain, participants are asked how many times they have written a piece of music, or created a mix tape, among other items. In the arts and crafts domain, participants are asked how many times they created an original decoration. In cooking, how many times they made up a new recipe. The visual and performing arts scale asks how many times participants painted a picture and performed in a play, respectively. All of these scales exhibited satisfactory reliability, with the Music Activities scale having the highest reliability, and the Literary Activities scale having the lowest reliability.

**Grit scale.** This 12-item self-report measure aims to tap participants' levels of a motivational construct termed _grit_ that includes both the consistency of participants' interest in and their perseverance on their long-term goals [66]. In previous work, Grit has been shown to predict educational and career outcomes across a variety of domains [36]. In its original conceptualization, this short measure contained two sub-scales: Consistency of Interest (e.g., _I often set a goal but later choose to pursue a different one_ [reverse coded]) and Perseverance of

Effort (e.g., *I finish whatever I begin*). Items on the Consistency of Interest sub-scale are all reverse-coded, while items on Perseverance of Effort scale are not. In this investigation, the individual sub-scales of this measure did not reach adequate reliability, so following with existing practice in the literature [67], the Grit Scale was scored unidimensionally, with all 12-items indicating a single underlying construct of Grit. Unidimensionally, the Grit Scale reached satisfactory reliability at both the composite (Cronbach's alpha) and factor (Omega, *H*) levels.

**Intolerance of uncertainty scale.** This 12-item scale is designed to tap participants psychological need for certainty, or their fear of the uncertain or unknown [68]. Intolerance of Uncertainty has been a key construct in current psychological understandings of worry, a phenomenon that is related to career choice [69]. This measure features two sub-scales: Prospective Anxiety (e.g., *I always want to know what the future has in store for me*), which is designed to tap how participants react to the possibility of an uncertain future, and Inhibitory Anxiety (e.g., *The smallest doubt can stop me from acting*), which focuses on the effect of uncertainty on participant behavior in the present. In this study, both of these scales exhibited satisfactory reliability and were scored separately, creating two scale-level scores for analysis.

**Emotional intelligence scale.** Broadly conceptualized as the ability to perceive, understand, and use emotions in oneself and in others [37], emotional intelligence has been measured in many ways in past literature, including performance tests [70] and self-report. Here, a short 12-item self-reported Emotional Intelligence measure [71] was administered. This measure features three scales: Self-emotional Appraisal (e.g., *I have a good understanding of my own emotions*), Others' emotional appraisal (e.g., *I am sensitive to the feelings and emotions of others*), and Uses of emotion (e.g., *I always encourage myself to try my best*). Each of these scales reached satisfactory reliability and were scores separately, creating 3 scale scores for future analysis.

**Big five aspects scale.** The Big Five Aspects Scale (BFAS; [38]) is a widely utilized self-report personality measure in which participants indicate levels of five principal aspects of personality, each of which is divided further into two facets. The "big five" dimensions of personality—Neuroticism, Agreeableness, Conscientiousness, Extraversion, and Openness—are all available on this measure. However, each of these principal five personality characteristics are further divided into two facets: Neuroticism contains both Volatility (e.g., *I get upset easily*) and Withdrawal (e.g., *I am afraid of many things*); Agreeableness contains Compassion (e.g., *I like to do things for others*) and Politeness (e.g., *I hate to seem pushy*); Conscientiousness is divided into both Industriousness (e.g., *I am not easily distracted*) and Orderliness (e.g., *I want everything to be 'just right'*); Extraversion has Enthusiasm (e.g., *I have a lot of fun*) and Assertiveness (e.g., *I know how to captivate people*); and Openness is divided in to Intellect (e.g., *I like to solve complex problems*) and Openness (e.g., *I need a creative outlet*). In this investigation each of the ten facets displayed strong reliability, and scoring therefore occurred at the facet-level, creating ten scores for later analysis.

**Procedures.** All participation in this study was conducted via the Internet with Qualtrics administration software. Informed consent was obtained before participants could move forward with the measures (these procedures were approved by the Institutional Review Board at the University of Denver). Study instructions asked participants to complete the measures with minimal distractions and recommended that they turn off electronic devices as well as close other websites or programs open on their computer. Because the AUT requires a significant amount of typing, participation required a traditional keyboard and participation via smartphone or tablet was not allowed. Participants were given two minutes to provide uses for each AUT item before they were automatically advanced to the next object, and they could not advance before those two minutes were up. After responding to all ten objects (i.e., after 20 minutes), participants were informed that the task was complete, and moved to the self-report

portion of the study. After all self-report measures were complete, participants responded to the demographic questions and logged out of the study website.

## Results and implications

Analysis of these data unfolded in two major phases. First, average levels of each of the constructs included in this study across the non-acting, student actor, and professional actor groups, in addition to significance tests for mean differences among those groups, are presented. Then, to provide a holistic picture of how these psychological attributes can, in concert, be used to describe the general attributes of actors, and even identify professional from student actors, a sequence of two Least Absolute Shrinkage and Selection Operator (LASSO; [72]) models are fit. These models are used to first identify those participants who are actors (either students or professionals) out of the entire sample, and then to further sort professional from student actors, when the comparison sample is left out of the analysis.

### Descriptive statistics and univariate mean comparisons

The measurement procedures used here produced 26 psychologically-relevant scores for each of the 296 participants in the dataset. These scores were produced via a CFA model on a standardized (z-score) metric, meaning that the grand mean across all participants was zero and standard deviation was one. However, means for the specific groups (i.e., non-actors, student actors, and professional actors) did exhibit statistically and practically significant differences. Please see Table 2 for means and standard deviations per group, as well as accompanying significance tests and effect sizes. At the $p < .05$ significance level, only three measurements did not exhibit significant mean differences (and the smallest effect-sizes): Politeness, Orderliness, and Uses of Emotion. Even at the much more conservative, Scheffé corrected significance level of $.05 \div 26 = .0019$, 17 out of the 26 constructs would still exhibit significant differences. In the sections that follow, each of these differences are presented, and immediate implications are briefly highlighted.

**Divergent thinking.** Fluency, Elaboration, and maximal Originality scoring of the AUT all showed significant differences among the groups in the following way: non-actors scored the lowest (below the grand mean), student actors scored in the middle (around the grand mean) and professional actors scored the highest (above the grand mean). However, the mean Originality scoring showed significant differences in a different pattern, in which the student actors scored the lowest, non-actors in the middle, and professional actors highest. All four of these measurements contribute to the general finding that professional actors have an average divergent thinking ability level that is heightened above non-actors and student actors, at least on a verbal task like the AUT. Of the dimensions of divergent thinking measured here, the strongest univariate effect size was found with Fluency.

**Creative activities.** The strongest effect-sizes present in these mean comparisons were found on the Inventory of Creative Activities: with the Performing Arts scale having the largest effect-size, followed by Music and Literary Activities. In addition, these three scales differed among the groups in a predictable way, in that the non-actors scored substantially below the grand mean on average, while both the student and professional actors scored above the grand mean on average. Interestingly, the student actors reported slightly more performing arts activities than the professional actors, perhaps because many opportunities for such activities exist in the college setting, and this scale did not differentiate among paid and un-paid activities. The cooking activities scale, somewhat counter to stereotype, indicated that undergraduate actors were engaged in substantially more cooking activities than were non-actors or professional actors.

**Grit.**   Following the pattern of much of the Inventory of Creative Activities and the AUT, the Grit Scale showed significant differences across groups with non-actors having the lowest, students being in the middle, and professional actors having the most Grit. Given the highly competitive nature of the acting profession, this finding may indicate that Grit supports professional actors as they transition from student to professional life.

**Intolerance of uncertainty.**   In contrast to this pattern, the Intolerance of Uncertainty scale showed the highest average scores for the non-actors (who were above the grand mean), indicating that they had the greatest need for certainty or fear of the unknown, while professional actors scored in the middle (around the grand mean), and student actors reported very low levels of Intolerance of Uncertainty (below the grand mean), or, worded a different way, high levels of tolerance of uncertainty. In line with previous findings about the difficulties faced by professional actors in their uncertain job market [1], this finding implies that professional actors, despite having a lower Intolerance of Uncertainty than non-acting adults, do not exhibit the very low levels of Intolerance of Uncertainty that student actors report. This pattern, in which student actors exhibit less Intolerance of Uncertainty than do professional actors, is perhaps not surprising given that they are situated within university contexts that are potentially more likely than the professional context to be personally supportive of individuals, and they are also much less likely than professional actors to be solely responsible for their livelihood (e.g., student actors may rely on financial aid or assistance from caregivers).

**Emotional intelligence.**   In a pattern that potentially runs counter to what would be expected given the emotional nature of Actors' work, professional actors reported an average level of Self-emotional Appraisal (a dimension of Emotional Intelligence) that was below the grand mean, non-actors reported an amount of Self-emotional Appraisal around the grand mean, and student actors reported substantially more than the grand mean. In contrast, professional actors reported an average level of Other-emotional appraisal that was around the grand mean, with student actors reporting substantially more than the grand mean and non-actors reporting substantially less. Taken together, these findings imply that actors' Emotional Intelligence (at least when self-reported) appears to be focused at others and not at themselves, although student actors did report heightened levels of both these facets of Emotional Intelligence, complicating that picture. Another potential explanation of this finding is that professional actors may be better calibrated in terms of their true levels of Emotional Intelligence, whereas non-actors and students may have exhibited a self-report bias that led them to overestimate their true levels of Emotional Intelligence, therefore moving their group means above the professional actors' group mean.

**Big five personality facets.**   As one of the only constructs included here that has been investigated previously within the literature on professional actors [12], the Big Five Personality traits are here examined with a high degree of specificity in that each of the Big 5 dimensions were quantified as two sub-dimensions or facets. In some cases (e.g., Extraversion), significant differences on one facet were in the opposite direction of the other facet, strongly suggesting the importance of facet-level measurement. Patterns of findings are discussed below.

*Openness/Intellect*. These two facets of personality are without a doubt the most widely studied in research on creativity and creative individuals and are currently considered the core of the creative personality [73]. However, patterns in significant mean differences across the three groups on these two facets were not the same. Specifically, both professional and student actors reported levels of Openness that was substantially above the grand mean, while non-actors reported levels of Openness that was substantially below the grand mean. The effect-size associated with the Openness significance test was the largest across any construct measured in this study, besides Creative Activities. This pattern implies that Openness, as a facet of

personality, is a strong discriminator of individuals who are or are not actors but does not differ across professional and student actors. The Intellect facet, on the other hand did differ among the student and professional actors, with student actors reporting much more Intellect than the professionals. In addition, the effect size associated with Intellect was much smaller than that associated with Openness, implying that Openness, more than Intellect, is a critical indicator of the acting personality.

*Extraversion.* Extraversion has long been associated with acting [8], and creative careers in general [74]. Here, significant group differences across non-actors, student actors, and professional actors were observed on both the Enthusiasm and Assertiveness facets of Extraversion. In both cases, the non-actors reported the lowest amounts (below the grand mean) and both the professional and student actors reported above the grand mean. Also in both facets, the student actors reported slightly more Extraversion than did the professionals, but the difference was more marked in Assertiveness, which displayed a larger effect size than did Enthusiasm.

*Conscientiousness.* In previous work (e.g., [75]), conscientiousness has been shown to be inversely related to creativity, and this study replicated that finding. Although the Orderliness facet of conscientiousness did not display significant differences, implying that the groups included in this study did not differ statistically on this attribute, the Industriousness facet did show significant differences with the non-actors reporting the greatest average amount of Industriousness (above the grand mean) and both the student and professional acting groups reporting substantially less Industriousness than the grand mean. This finding suggests that previous evidence about Conscientiousness and its inverse relation to creative work may have been driven by the Industriousness facet. As a possibility, the Industriousness facet may be specifically inversely related to mind-wandering activities that have been shown to support creative insight [76].

*Agreeableness.* Although no significant differences were found on the Politeness facet of Agreeableness, significant differences were found on Compassion. Specifically, non-actors reported the lowest average levels of Compassion (below the grand mean), professional actors reported a level of Compassion that was right around the grand mean, and student actors reported the highest levels of Compassion, substantially higher than the grand mean. This finding suggests that actors in general have heightened levels of Compassion, but that pattern is most marked with student actors, who report more Compassion than do professionals.

*Neuroticism.* Both the Volatility and Withdrawal facets of Neuroticism displayed significant differences across the groups, and the differences on both those facets were in the same direction. In particular, the non-actors displayed the lowest levels of Neuroticism (below the grand mean), student actors reported levels of both facets of Neuroticism that was around the grand mean, and professional actors reported the highest levels of Neuroticism (above the grand mean). This finding is in line with previous work (e.g. [12]) that observed heightened levels of Neuroticism with professional actors, as well as other classic work (e.g. [8]) that argued that this personality difference was a key to understanding the differences among student and professional actors.

## LASSO classification models

Although the univariate mean comparisons presented in the previous section are relevant to understanding the psychology of professional actors, they fall short of providing a full picture of how actors—whether student or professional—holistically differ from non-actors, and how professional actors are distinguished from students psychologically. In effect, the univariate tests are useful in that they each isolate a single psychological construct and elucidate group

differences on that construct. But, as has been understood in multivariate psychology for many years (e.g. [77]), univariate comparisons among groups may mask important differences that can emerge when several constructs are analyzed simultaneously. There are a variety of existing analytic strategies to find substantively meaningful patterns in high-dimensional multivariate data like that used here (see [78]). One of those strategies is to reverse the predictive direction of the general linear models used above—where the categorical group indicator was the predictor, and the continuous measurement was the outcome—and attempt to use the 26 psychological attributes measured in this study to effectively sort the participants back into their groups (i.e., continuous attributes will be predictors, categorical groups will be the outcome).

In order to accomplish this conceptual and analytic strategy, there are a number of available statistical methodologies. For example, discriminant function analysis has historically been, and remains useful in psychology to classify participants (e.g., [79]), and logistic regression (as well as its ordinal and multinomial versions) has also been a fruitfully applied model (e.g., [80]). However, these traditional statistical methods do have a number of limitations worth considering here. For one, these methods require an a-priori specified list of predictor variables that enter the model in a specific order: decisions that researchers sometimes do not have the theoretical knowledge to make, and that can have hidden influences on the strength and even direction of the predictive relations from the predictors to the outcome [81]. In addition, these models are frequently over-fit, meaning that their predictive capacity (e.g., R-square) is almost always strongest in the dataset to which they are originally fit, and are weaker in validation or replication datasets; a phenomenon known as shrinkage [82].

A more modern alternative to these methods is LASSO, which estimates a penalization parameter ($\lambda$), that is applied to the model parameters in order to minimize its degree of over-fitting and therefore minimize future shrinkage of the coefficients should the model be fit to a different dataset [72, 83]. The larger the $\lambda$ of a LASSO model, the more penalized (i.e., reduced) the LASSO coefficients will be, relative to a traditional method. In addition, LASSO algorithmically enters the predictor variables into the model in order to determine which predictors are necessary to retain (i.e., are significant predictors), and which predictors can be dropped from the model. LASSO models can also be specifically cross-validated by folding the analytic dataset a certain number of times and estimating a model that minimizes over-fitting and future shrinkage as well as selecting maximally important predictors for classification of participants [43]. In addition, a logistic version of LASSO, in which the outcome variables are categorical, has been meaningfully used in the psychological literature as a classification model (e.g. [84, 85]). Given the advantages of LASSO, it was applied here to test the capacity of the 26 psychological constructs measured in this study to discriminate actors (either students or professionals) from non-actors, and to sort professional and student actors from one another.

In interpreting the results of the LASSO models in this study, and especially in interpreting how the results of the LASSO models differ from the results of the univariate tests above, it is important to consider that the LASSO coefficients are meant to represent the predictive capacity of that variable *with all other variables in the model controlled for*. The results of the LASSO models below can be substantively different than the univariate results above for this reason. For instance, it is possible (and even likely) that some variables that exhibited univariate differences above will be dropped from the LASSO models due to their lack of predictive power over and above the other included variables. The inverse is also possible: just because a variable did not display univariate differences above does not mean that it is not a useful predictor of acting, when combined with the other variables in the model. For these methodological reasons, we would argue that the LASSO coefficients calculated here are a more true-to-life depiction of how professional and student actors differ from non-actors, or how professional and

student actors differ from each other. This is because the various psychological attributes studied here do not, in truth, operate in an isolated or univariate way. Instead, these psychological attributes operate in conjunction with one another in order to support individuals in their creative development and profession.

**Identifying actors.** Using all 296 participants included in this study, a cross-validated LASSO model was fit using all 26 measured constructs as potential predictors to sort non-actors (n = 92, coded 0) from actors (professional and student combined n = 204, coded 1). The model was cross-validated on ten folds of the dataset. The unpenalized model attained an R-square of .68 (equivalent to maximum likelihood logistic regression), but the optimal λ parameter estimated by the model was .01, and when applied to the model coefficients, reduced the R-square to .63. This LASSO model produced a predicted probability of being an actor for every participant in the dataset. If that predicted probability was over .50, they were considered to have been sorted into the actors' group, below .50 they were considered to have been sorted into the non-actors' group. Please see Table 3 for full classification accuracy information on these models. The LASSO model was capable of correctly identifying 96.57% (n = 197) of the actors in these data, correctly identifying 81.53% (n = 75) of the non-actors, and had a total classification accuracy of 91.9%.

When the cross-validated predictive coefficients were estimated using the optimal λ penalization, 17 out of 26 predictors were selected by the model. The LASSO predictive coefficients for these 17 constructs, as well as their coefficients from a post-LASSO logistic regression run only with these 17 predictors, are available in Table 4. Fitting with expectations, the Performing Arts Creative Activities measure was the most important measure in sorting actors from non-actors. The Industriousness construct was the second most strongly weighted attribute in the model, although it was weighted in a negative direction, implying that a low-level of Industriousness is a strong indicator that a participant is an actor in this dataset. This finding may be interesting going forward in the field of creativity research because it implies that the creative personality is not marked by high levels of Conscientiousness, especially the Industriousness facet. Recent neurological work suggests that Industriousness is a personality facet that may moderate the brain's activities when thinking divergently [86], suggesting it is a highly relevant personality facet for future research. The Musical Creative Activities measure was also strongly weighted by the model, followed by Assertiveness, which was weighted in the positive direction (i.e., actors are identified by higher levels of Assertiveness). Both of these

**Table 3. Classification tables from LASSO models.**

| Sorting Actors from Non-Actors (Model 1) | | | |
|---|---|---|---|
| True Category | Sorted Category | | Total |
| | Non-Actor | Actor | |
| Non-Actor | 75 | 17 | 92 |
| Actor | 7 | 97 | 104 |
| Total | 82 | 214 | 296 |
| Sorting Professional from Student Actors (Model 2) | | | |
| True Category | Sorted Category | | Total |
| | Student | Professional | |
| Student | 63 | 37 | 100 |
| Professional | 29 | 75 | 104 |
| Total | 92 | 112 | 204 |

*Note*: Model 1 percentage correctly sorted was 91.9%, Model 2 percentage correctly sorted was 67.7%

**Table 4. Predictive coefficients for LASSO models.**

| Psychological Predictor | Sorting Actors from Non-Actors | | Sorting Professional from Student Actors | |
|---|---|---|---|---|
| | Logistic LASSO | Post-LASSO Logit | Logistic LASSO | Post-LASSO Logit |
| Fluency | 0.247 | 0.098 | -- | -- |
| Elaboration | 0.338 | 0.476 | -- | -- |
| Originality (mean) | -- | -- | 0.282 | 0.460 |
| Originality (max) | 0.232 | 0.442 | -- | -- |
| Literary Activities | 0.156 | 0.124 | 0.341 | 0.972 |
| Musical Activities | 0.817 | 1.118 | -- | -- |
| Crafting Activities | 0.242 | 0.503 | -- | -- |
| Cooking Activities | -0.239 | -0.599 | -0.289 | -0.389 |
| Visual Art Activities | -- | -- | -- | -- |
| Performing Arts Activities | 1.23 | 1.609 | -- | -- |
| Grit | -- | -- | -- | -- |
| Intolerance of Uncertainty (Prospective) | -0.194 | -0.569 | -- | -- |
| Intolerance of Uncertainty (Inhibitory) | -- | -- | 0.145 | 0.072 |
| Self-emotional Appraisal | -- | -- | -- | -- |
| Other-emotional Appraisal | -- | -- | -0.126 | -0.018 |
| Uses of Emotion | 0.114 | 0.173 | -- | -- |
| Openness | 0.291 | 0.366 | -- | -- |
| Intellect | -0.059 | -0.426 | -- | -- |
| Enthusiasm | -- | -- | -- | -- |
| Assertiveness | 0.507 | 0.896 | -- | -- |
| Industriousness | -0.844 | -1.388 | -- | -- |
| Orderliness | 0.212 | 0.776 | -- | -- |
| Compassion | 0.147 | 0.336 | -- | -- |
| Politeness | -- | -- | -- | -- |
| Volatility | -- | -- | 0.029 | 0.297 |
| Withdrawal | -- | -- | -- | -- |
| Constant | 1.651 | 2.083 | -0.001 | -0.891 |

positively weighted constructs fit with expectations, which generally consider Musical activities adjacent to acting, and the Assertiveness facet of Extraversion central to the personality of a performing artist.

Three out of the four measured divergent thinking scores (i.e., Fluency, Elaboration, and maximum Originality, but not mean Originality) were selected by the model and weighted in the positive direction, implying that high levels of divergent thinking ability can be used to sort actors from non-actors. Of these three divergent thinking attributes, Elaboration was the most strongly weighted, suggesting that actors' ability to expound on their ideas distinguishes them from non-actors: a finding that appears to fit with the highly verbal demands of the acting profession. The Prospective Intolerance of Uncertainty scale was included in the model and weighted negatively, indicating that actors have less fear of an unknown future than do non-actors: a personality difference that may allow actors to survive in an unstable and under-resourced profession. Also, despite being a non-significant comparison in the univariate analysis above, the Uses of Emotion scale of the Emotional Intelligence measure was selected for the model and weighted in the positive direction, which makes theoretical sense given the need to use emotions during the acting process.

The personality facets of Openness, Orderliness, and Compassion were also selected for the model and weighted positively, while the Intellect personality facet was selected for the model

but weighted negatively. Both the findings related to Openness and Compassion fit with general expectations reviewed here, in that actors would be expected to be generally open to new experiences, and compassionate to others, in order to develop the characters they are called upon to embody and perform. The positive weighting of the Orderliness facet of the Conscientiousness dimension was perhaps less expected, especially given the negative weighting of Industriousness. Taken together, these findings imply that actors generally prefer to follow a set schedule and prefer tidiness above disorder (positive weighting of Orderliness); while actors also struggle to carry out plans, make major decisions, and resist being distracted (negative weighting of Industriousness). In our view, this set of findings highlights a general personality that may be able to simultaneously support the rigorous demands of a rehearsal schedule, while also allowing for the mind-wandering cognitive style that has long been associated with original thought [76]. The Cooking scale of the Creative Activities Inventory was also weighted negatively, suggesting that non-actors are more likely to express their creativity through cooking than are actors: a finding that may suggest a preference on the part of actors for eating-out or eating at work during rehearsals and performances.

**Identifying professionals.** Then, in order to more specifically analyze salient psychological patterns among student and professional actors, the sample was restricted to include only those participants who were actors, with student actors ($n = 100$) coded as 0 and professional actors ($n = 104$) coded as 1. A LASSO model with the same 26 potential predictors was fit to these data, and cross-validated with ten folds of the dataset. The psychological differences among student and professional actors were more subtle than the differences among non-actors and actors, and the unpenalized model reached an R-square of .24. The optimal $\lambda$ parameter was estimated to be .04, which corrected the penalized R-square down to .22. This cross-validated, penalized LASSO model selected 6 predictors from the initial 26 to retain.

Based on this LASSO model's predicted probability of each participant being a professional actor (following the same $>|<$ .50 rule as above), classification information for this model is available in Table 3. As can be seen, the model was capable of correctly identifying 72.11% of the professional actors (n = 75), and 63.00% (n = 63) of the student actors, with a total classification accuracy of 67.7%. The LASSO predictive coefficients for the 6 included constructs, as well as their coefficients from a post-LASSO logistic regression run only with these 6 predictors, are available in Table 4.

Age is the most obvious difference between professional and student actors. When Age is included as a predictor in this cross-validated LASSO model, the unpenalized R-square is much higher (.82). The optimal $\lambda$ is then estimated as 2.83, correcting that R-square down to .76. This penalized and cross-validated LASSO model also achieves a 95.6% classification rate. However, given the focus in this investigation on *psychological* attributes of professional actors, we decided to present and interpret the model with Age (and other demographic variables) not included.

The most strongly weighted predictor in the psychologically focused LASSO model (with the 67.7% classification accuracy) was the Literary scale of the Creative Activities Inventory, which was weighted in the positive direction indicating that a higher degree of participation in literary activities (e.g., writing stories or scripts) was a mark of a professional actor in this dataset. In contrast, the Cooking scale of the Creative Activities Inventory was weighted in a negative direction by the model, suggesting that professional actors apply their creativity to cooking significantly less than do the student actors in these data. Despite not being selected for the LASSO model that sorted actors from non-actors, the mean Originality predictor was selected to sort professional from student actors and was weighted in the positive direction. Combined with the coefficient associated with maximal Originality in the earlier LASSO model, this indicates that actors in general are distinguished by the production of at least one

highly original idea per AUT prompt, but professional actors can be identified as those who produce ideas that are more Original on average.

The Inhibitory Anxiety scale of the Intolerance of Uncertainty measure was selected for this model and weighted in the positive direction, implying that professional actors reported more inhibition in response to uncertainty than did student actors, who were very tolerant of uncertainty: a finding that appears to fit with the widely held conceptualization of student actors' lifestyle as relatively stable and well-supported (either by caregivers or financial aid), and professional actors' lifestyle as unstable and under-resourced. The Other-emotional Appraisal scale of the Emotional Intelligence measure was also selected for this model and weighted negatively, indicating that professional actors reported less ability to perceive the emotions of others than did undergraduate acting majors. As previously discussed in relation to the univariate findings above, the greater Emotional Intelligence of student actors than professional actors could potentially be explained by a self-report bias in students and better calibration to truth in professionals; or potentially it could relate to a need within the professional acting community to lessen emotional awareness, given their difficult and economically unstable lifestyle, in comparison to the relatively stable lifestyle of students.

Evidence from social neuroscience research may also relate to this finding of greater Other-emotional Appraisal in student actors. It is understood within the social neuroscience literature that the neurological system responsible for seeking rewards is highly influenced by the social rewards available from peers, and that this system develops relatively early in adolescence; in contrast, the neurological control system responsible for self-management is likely not fully developed until an individual's mid-20s [87, 88]. Therefore, it may be that undergraduate actors are closely emotionally attuned to their peers in their theater productions given their as-yet-not-fully-developed neurological system, while professional actors have the adult neurological capacity to control or ignore their emotional awareness when it is necessary. Finally, the Volatility facet of Neuroticism was selected for this model and weighted in the positive direction, suggesting that professional actors reported being more prone to angry or upset moods than were student actors: another specific finding that, in our view, likely relates to the socially difficult and under-resourced aspects of the acting profession.

## Discussion

### Key findings

This study has been the first to administer a wide-array of psychological measures (including both creativity-related and personality attributes) to professional and student actors, as well as a non-acting comparison group. As such this study has a number of findings to bring to the current understanding of the psychology of actors and acting. Some very specific implications of our findings are pointed out in the Results section above, and the overarching patterns that require greater emphasis and detail are presented here, in the Discussion section, as Key Findings.

**Actors display heightened divergent thinking ability.**   In line with hypotheses, both student and professional actors displayed significantly higher levels of DT than did non-actors. However, the patterns of DT-related findings were nuanced in terms of which dimensions of DT were most strongly associated with status as an actor, or as a professional actor specifically. For instance, three dimensions of DT: Fluency, Elaboration, and maximal Originality were selected by the LASSO model to classify actors from non-actors, and among those three dimensions, Elaboration was the most strongly weighted. This finding suggests that is an individual's ability to flesh-out or explain their ideas—even when their quantity of ideas (Fluency) or maximal level of the novelty of those ideas is statistically controlled—that most distinguishes

actors from non-actors. Further, although maximal Originality was selected as a significant predictor by the LASSO model that classified actors and non-actors, the mean Originality of AUT responses was selected by the LASSO model that classified student and professional actors. In our view, this finding is a key contribution of the current study, in that it suggests (and is the first to do so as far as we are aware) that participation in acting seems to require individuals to be capable of generating at least one highly Original idea within an allotted period of time, but professional success as an actor appears to specifically depend on an individual's capability to produce more original ideas on average, across all of the ideas they generate. Such a finding may be driven by the continuous demands on professional actors to generate relatively novel suggestions or ideas, and therefore professionals may be required to be Original thinkers on average across all of their generated ideas, while students are required to generate a smaller number of Original ideas in their university theater work. This finding is in line with research into other areas of expertise (e.g., medicine; [89]) that suggests that a distinguishing attribute of experts is the capability to produce high-quality work regularly and across contexts, while students are more likely to produce very high-quality work sporadically.

**Assertiveness and openness, not enthusiasm or intellect, identifies actors.** In past work on creativity in general, and actors and acting specifically [12, 73] the personality dimensions of Openness to Experience and Extraversion have been identified as two dimensions on which actors would be expected to be high. However, by undertaking this analysis at the more finely-grained personality facet level, this work showed that only one facet of each of these two larger factors was positively predictive of an individual being an actor. Specifically, only the Openness facet of the larger Openness to Experience factor (that also includes Intellect), and only the Assertiveness facet of the Extraversion factor (that also included Enthusiasm) were positive predictors. In addition, the Intellect facet of the Openness to Experience factor was actually a negative predictor selected by the LASSO model to classify actors, while the Enthusiasm facet was simply dropped by the algorithm. The strong weighting of the Assertiveness facet, rather than the Enthusiasm facet of Extraversion appears to make sense in this context, in that Assertiveness is indicated by items such as "*I know how to captivate people*", that seem to correspond to actors' work well.

**Professional actors are identified by their volatility.** As discussed previously, when demographic variables such as Age were entered into the LASSO model, professional actors and student actors were easily distinguished. However, without demographic differences in the model, the psychological differences between professional and student actors were subtler in these data than were the differences between actors and non-actors. One predictor that was utilized by the LASSO model to classify professional actors was their Volatility. This finding is closely in line with previous work (e.g., [8, 40]) that suggests professional actors can be distinguished from non-actors or student actors by their high-levels of negative personality traits. Indicators of Volatility administered here included items such as "*I get upset easily*" and "*I change my mood a lot*", and, based on the overarching patterns in the univariate comparisons and LASSO models conducted in this study, this emotional Volatility trait is an identifying attribute of professional actors. As suggested in previous work [10], professional actors are specifically at risk for threats to their emotional well-being given the very high economic riskiness of their profession, coupled with the high emotional vulnerability that is required of an actor to do their work, which may contribute to them developing a highly emotionally volatile personality.

In contrast to the highly volatile environment within which professional actors are situated, student actors appear much more likely to be working in stable and supportive environments (e.g., a conservatory setting), and may not be solely responsible for their own financial well-being, relying instead on parental support or student loans. This major difference in

environment may explain why professional actors displayed higher levels of Volatility in their personality than did students. Another potentially influential factor in the development of Volatility may be age and natural development, but the cross-sectional methodology utilized here was unable to disentangle to effects of aging and development, on one hand, and systematic changes in environmental stability, on the other. Future longitudinal work would be needed in order to address these open questions. As can be seen in Footnote 1 of this paper, when age was included in the LASSO model that sorted professional from student actors, the classification rate was much stronger (i.e., 96%). Such a finding implies that age is the clearest difference between professional and student actors, but in this investigation, we chose to report and interpret LASSO models without age, because the focus of this work was on psychological, not demographic, attributes.

## Limitations

Like any research in the social sciences, this study has a number of limitations and delimitations that should be considered when generalizing its findings. For example, one measurement-related limitation of this study was our sole reliance on verbal—rather than figural—DT tasks. In our view, the focus on verbal DT was a way to tailor the measurement procedures of this study to the highly verbal work of professional and student actors, but it remains a future direction to uncover whether statistically or practically significant patterns could be uncovered using figural DT tasks (which typically require drawing). In addition, this research utilized three specific groups of participants that differed on their acting status (i.e., professionals, students, and nonactors) in order to make inferences about what distinguished professional and student actors from each other and from the general population. However, to fully understand the trajectory of expertise development within the acting profession, longitudinal data collection in which student actors are followed during their transition to professional life would be needed. This longitudinal work may be particularly difficult and costly given the very low proportion of undergraduate acting students who persist in the profession after graduation: an attribute of artistic career development that distinguishes it from other areas of education and expertise. Finally, it is possible that student or professional actors that tend to book higher- or lower-profile roles (e.g., lead roles versus ensemble roles) also differ systematically in their psychological attributes. In this study, we did not ascertain among the student or professional actors which individuals in the sample were being cast in more central roles, and this appears to be a fertile ground for future research that drills deeper into the psychology of student and professional actors.

## Conclusion

For those of us who are inspired and moved by the performing arts, the work of the professional actor is a powerful and important art that deserves to be strongly valued by society. In addition, it is understood that individuals differ psychologically from one another, and those differences are at least in part driven by their professional or educational contexts, and concomitantly that those psychological differences among individuals can contribute greatly to professional or educational success. This study has specifically endeavored to increase what is known about the psychological individual differences that distinguish professional actors. In the future, these results may potentially be useful not only for the psychological research community but also for actors themselves, and especially for educators that train undergraduate level student actors, or organizations of professional actors (e.g., Actor's Equity) who seek to understand the needs of their members. As such, this study contributes to what we see as a critically important endeavor for psychology and the social sciences: a deep understanding of the mental attributes that play a role in every area of the human experience.

## Acknowledgments

The authors would like to thank Dr. Paula Thomson and one other anonymous expert reviewer for their very insightful comments on this manuscript during the review process.

## Author Contributions

**Conceptualization:** Denis Dumas, Michael Doherty.

**Data curation:** Michael Doherty, Peter Organisciak.

**Formal analysis:** Denis Dumas, Peter Organisciak.

**Methodology:** Denis Dumas, Peter Organisciak.

**Project administration:** Denis Dumas, Michael Doherty.

**Software:** Denis Dumas, Peter Organisciak.

**Writing – original draft:** Denis Dumas.

**Writing – review & editing:** Denis Dumas, Michael Doherty, Peter Organisciak.

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
