## [Decision Letter · Decision Letter 0]

27 May 2020

PONE-D-20-02455

The Psychology of Professional and Student Actors: Creativity, Personality, and Motivation

PLOS ONE

Dear Dr. Dumas,

Thank you for submitting your manuscript to PLOS ONE. After careful consideration, we feel that it has merit but does not fully meet PLOS ONE’s publication criteria as it currently stands. Therefore, we invite you to submit a revised version of the manuscript that addresses the points raised during the review process.

We look forward to receiving your revised manuscript.

Kind regards,

Paolo Roma

Academic Editor

PLOS ONE

**Journal Requirements:**

4. Thank you for stating the following in the Financial Disclosure section:

'The author(s) received no specific funding for this work.'

We note that one or more of the authors are employed by a commercial company: 'Actor’s Equity Association'.

Please respond by return email with an updated Funding Statement and Competing Interests Statement and we will change the online submission form on your behalf.

**Comments to the Author**

1. Is the manuscript technically sound, and do the data support the conclusions?

Reviewer #1: Yes

Reviewer #2: Yes

2. Has the statistical analysis been performed appropriately and rigorously? 

Reviewer #1: Yes

Reviewer #2: Yes

3. Have the authors made all data underlying the findings in their manuscript fully available?

Reviewer #1: Yes

Reviewer #2: No

4. Is the manuscript presented in an intelligible fashion and written in standard English?

Reviewer #1: Yes

Reviewer #2: Yes

5. Review Comments to the Author

Reviewer #1: PlosOne Review – March 15, 2020

Manuscript ID - PONE-D-20-02455

Title: The Psychology of Professional and Student Actors: Creativity, Personality, and Motivation

Main Findings and Significance to the Discipline

The main findings of this study include personality traits, divergent thinking responses, creative activities, emotional intelligence and grit. The study demonstrated factors that differentiate professional actors, from student actors and non-actors. The statistical analyses were novel and strengthened the reporting of the findings. Because actors are under-researched, this study is important and necessary.

General Comments:

In general, the manuscript is very well written and clearly organized. The authors have carefully outlined the study protocol and methods of analysis. They have situated their findings within the larger field of research and their findings are important. The authors have included many important references that support the purpose of their study and the importance of their findings. They have also provided enough information so that the study can be replicated and provided information about how to access their original data. It is recommended that the authors also include work by Zorana Ivcevic and colleagues as well as Thalia Goldstein and Ellen Winner. They have all conducted work on emotional intelligence, theory of mind, and empathy in actors and their relationship to creativity. Lastly, a careful edit will catch inconsistencies in citation formatting and reference formatting.

Abstract:

The abstract is well organized and concise. It would be helpful to include the number of actors, student actors and non-actors in the abstract as well.

Review of Literature:

This section is generally well organized. Providing an historical overview positions the current study and highlights the importance of this study. In the discussion on empathy the authors should include research findings by Winner and Goldstein. They also examined empathy and theory of mind in actors.

Methods:

This section is detailed and situates this study so that it could be easily replicated. Can the authors provide more information on the control sample? The control group also clearly engages in creative activities given their responses. The measures and the statistical analyses were detailed.

Results:

In combination with the tables and figure, the results were fully explained. The authors also offer solid reasons for their statistical choices. The summary results for each measure was very helpful.

Discussion and Conclusions:

The overall findings were summarized succinctly. The limitations were relevant. As stated by the authors, longitudinal work is needed to understand the large attrition rate as student actors move into a professional career. It would be helpful to provide a stronger statement differentiating student actors from professional actors, especially because student actors who are trained in a conservatory setting are working in stable environments, hence their willingness to take risks would be higher. Age further influences some of these differences, a factor that was minimally addressed in this study.

References:

A careful edit is required to correct inconsistencies in formatting.

Tables and Figures:

The tables and figure are helpful and clearly outlined.

Specific Comments:

P. 3 – change (Kogan, 2002) to the correct formatting.

P. 5 – Jague – should be spelled Jaque to match the reference listing

P. 12 – change doctoral agree to doctoral degree

p. 15 – Likert is incorrectly spelled

p. 31 – there is a double wording of “that suggests that suggests”

Reviewer #2: Comments about above questions:

Data analysis- analytic procedure is well detailed with citations. The authors nicely compare it to more traditional procedures that most researchers would be able to easily understand. However, I am not familiar with LASSO specifically, and so cannot determine the fullest accuracy of their analysis.

Data Availability- The authors note in their submission information that data is available on their site, however this not mentioned in the actual manuscript.

Writing quality - minimal typos present. These should be easily found by the authors in their revision process.

Revise and Resubmit

The manuscript, The Psychology of Professional and Student Actors: Creativity, Personality, and Motivation, is an interesting paper describing a newer approach to identifying group membership among actors and non-actors using a novel set of measures of creativity and other psychological dimensions. Although it has many strengths, including the LASSO approach, there are some suggestions that the authors might address to strengthen the paper.

Major revisions

1. The “summary of extant work” section of the introduction could be streamlined. For instance, it begins as if the research is presented chronologically (e.g., “earliest work”), but then goes into negative psychological attributes, cognitive and expertise development, followed by a final paragraph that overlaps with prior content. I think taking the final paragraph in that section and incorporating it into other sections of the paper would improve flow (e.g., paragraph of Nettle’s personality and ASD components could be incorporated into the first paragraph of section or the paragraph on negative emotional attributes. Also, worth possibly including is the recent Gentzler, DeLong, and Smart (2019) for their comparisons of student acting majors and non-actors on a variety of psychological dimensions. For instance, they assessed temperament which can be regarded as similar to personality.

2. In the divergent thinking assessment section, I thought the points made by the authors as to why that measure could be a useful indicator of performance for actors was good. However, I think there is much to be elaborated. First, has that measure actually been assessed among actors? It’s lack of use for predictive power for determining group membership is made clear, but if it has been assessed as an outcome among participants is unclear as written.

3. More detail about the personality facets could be included. It’s unclear if the authors’ comments about those that load onto the Big 5 is in reference to the emotional intelligence and grit previously mentioned, or if they are referring to other facets. Given that the Big5 dimensions are often broken down to the facet-level, more clarity is warranted.

4. A clear statement about the main benefits of the project is warranted in the introduction. The first time the authors describe the benefits concretely, is in the final paragraph of the paper. The authors begin the manuscript with noting the research questions that stem from comparing the unique lifestyles of actors to non-actors, but it’s unstated as to why that matters. Although I agree it’s a great research question quite interesting to study, briefly mentioning the benefits of a study like this would greatly strengthen the selling point of the paper.

5. Regarding the undergraduate acting majors, what year in the program were they? Do you have any indicators of how much experience those students had on stage (e.g., number of productions performed in)? I would think that those who participated in more productions would be better actors compared to less experienced students and may have important implications for the other qualities measured. Also, was there an index to confirm the non-acting participants had not performed in any productions before?

a. Also, it would help to know if there was any difference in the students who might be “theatre” vs “acting” students. For example, could there be students with different training/ coursework (e.g., vocal musical training for musical theatre vs. vocal training for speech alone), creating subgroups within the major? If there are differences, it would be a nice addition to know if there are any differences in the students’ trainings and experience and any implications of that. If there are no differences and “theatre and acting” is really the same program at U of Denver, it would be useful to simply adjust the wording in the paper to indicate that’s just what the program is called.

6. I thought that the description of the LASSO technique in the results section was thorough and easy to follow. I found it interesting that some variables that showed no univariate differences between groups were selected for the LASSO model predicting actor status. I think some elaboration is needed to explain why that happened, and its implications in future research for statistical methods and for understanding actors from non-actors.

7. For some findings, descriptive implications are missing from the results or discussion (e.g., emotional intelligence measures, tolerance of uncertainty). For instance, a description of implication regarding the unexpected finding at the univariate level for emotional intelligence and the more anticipated finding with LASSO is warranted. Additionally, the implications regarding findings with tolerance of uncertainty are not elaborate. For example, the finding that students are more tolerant of uncertainty than professionals is not surprising to me given that students are less likely to be solely responsible for their own livelihood (e.g., receiving financial aid or assistance from caregivers while in college). Adding brief explanations for why the separate dimensions of emotional intelligence, personality, etc. were found would be very useful.

Minor revisions

1. A few typos are apparent (e.g., missing punctuation after citation brackets, missing “s” for the plural form of the word “responses”, spelling of “facets” as “facts”).

6. PLOS authors have the option to publish the peer review history of their article (what does this mean?). If published, this will include your full peer review and any attached files.

Reviewer #1: Yes: Paula Thomson

Reviewer #2: No

---

## [Author Response · Author response to Decision Letter 0]

18 Jun 2020

***All Reviewer Responses are also available in the cover letter to this manuscript****

Editor’s comments:

Thank you for submitting your manuscript to PLOS ONE. After careful consideration, we feel that it has merit but does not fully meet PLOS ONE’s publication criteria as it currently stands. Therefore, we invite you to submit a revised version of the manuscript that addresses the points raised during the review process.

As part of this revision, we have carefully reviewed the PLOS ONE style requirements—including those for file naming—and have conformed to these requirements in our revised re-submission. 

Thank you very much for reminding us about this issue. We prefer to wait until this paper is accepted to make the data publicly available and will be happy to upload the data to our website upon acceptance. Our research group’s website currently has a number of open access functions and we plan on adding a “data” tab when this paper is accepted, and will upload the data to that tab. Our lab’s website can be found here: https://openscoring.du.edu/. 

To make this issue plain to readers, and in response to Reviewer 2’s comment #22 below, we have now inserted a sentence on p. 12 of the manuscript immediately under the “Methodology” heading, describing how to access these data for readers and linking to the website. 

Thank you for reminding us about this issue. I have now linked my ORCID account to the PLOS submission portal. My ORCID iD is: https://orcid.org/0000-0002-8446-4720

4. Thank you for stating the following in the Financial Disclosure section:

'The author(s) received no specific funding for this work.' We note that one or more of the authors are employed by a commercial company: 'Actor’s Equity Association'.

Please provide an amended Funding Statement declaring this commercial affiliation, as well as a statement regarding the Role of Funders in your study. If the funding organization did not play a role in the study design, data collection and analysis, decision to publish, or preparation of the manuscript and only provided financial support in the form of authors' salaries and/or research materials, please review your statements relating to the author contributions, and ensure you have specifically and accurately indicated the role(s) that these authors had in your study. You can update author roles in the Author Contributions section of the online submission form. Please also include the following statement within your amended Funding Statement.

Thanks for pointing out this important issue that we needed to explain more thoroughly. The second author of this study (Michael Doherty) is a professional actor who worked with the scientific team on this study to formulate the research questions, select measures to administer, and manage the data collection. The author contributed professional expertise, not at the behest of the professional organization, and we believe the interdisciplinary nature of this work is a strength. 

As a professional stage actor, Michael’s employment is typically at a number of theater companies and playhouses throughout any given year. In the United States, Actor’s Equity is the labor union that represents stage actors—of which Mike is a member—and that union represents the interests of actors across all theater companies where they work, and provides their key benefits (e.g., health insurance, retirement). Therefore, we found that listing Mike’s affiliation as “Actor’s Equity” most reasonably communicated his involvement in the profession. 

Actor’s Equity did not provide any financial support for this study and did not directly provide salary to the second author either. In the context of the United States, union membership is absolutely crucial to the careers of stage actors, so we felt that listing Actor’s Equity as Mike’s affiliation was warranted and communicated his professional stature to readers. 

5. Please also provide an updated Competing Interests Statement declaring this commercial affiliation along with any other relevant declarations relating to employment, consultancy, patents, products in development, or marketed products, etc. Within your Competing Interests Statement, please confirm that this commercial affiliation does not alter your adherence to all PLOS ONE policies on sharing data and materials by including the following statement: "This does not alter our adherence to PLOS ONE policies on sharing data and materials.” (as detailed online in our guide for authors http://journals.plos.org/plosone/s/competing-interests). If this adherence statement is not accurate and there are restrictions on sharing of data and/or materials, please state these. Please note that we cannot proceed with consideration of your article until this information has been declared.

Please respond by return email with an updated Funding Statement and Competing Interests Statement and we will change the online submission form on your behalf. Please know it is PLOS ONE policy for corresponding authors to declare, on behalf of all authors, all potential competing interests for the purposes of transparency. PLOS defines a competing interest as anything that interferes with, or could reasonably be perceived as interfering with, the full and objective presentation, peer review, editorial decision-making, or publication of research or non-research articles submitted to one of the journals. Competing interests can be financial or non-financial, professional, or personal. Competing interests can arise in relationship to an organization or another person. Please follow this link to our website for more details on competing interests: http://journals.plos.org/plosone/s/competing-interests

Michael Doherty’s affiliation with Actor’s Equity Association did not and does not alter our adherence to PLOS ONE policies on sharing data and material. Actor’s Equity Association did not provide any financial support for this study, including in the form of author salary. As a collective authorship team, we have no competing interests that could interfere with the scientific process related to this work. 

 Reviewer 1 (Paula Thomson) Comments: 

6. The main findings of this study include personality traits, divergent thinking responses, creative activities, emotional intelligence and grit. The study demonstrated factors that differentiate professional actors, from student actors and non-actors. The statistical analyses were novel and strengthened the reporting of the findings. Because actors are under-researched, this study is important and necessary. In general, the manuscript is very well written and clearly organized. The authors have carefully outlined the study protocol and methods of analysis. They have situated their findings within the larger field of research and their findings are important. The authors have included many important references that support the purpose of their study and the importance of their findings. They have also provided enough information so that the study can be replicated and provided information about how to access their original data. 

Thank you very much Paula for your generally positive appraisal of our work with this manuscript. I remember hearing your talk about creativity research with performing artists in Oregon, perhaps 2 years ago, which was around the time we were first conceptualizing this study. I am very happy that you were able to review this manuscript and feel that, with the incorporation of your edits, this manuscript is much stronger now. 

7. It is recommended that the authors also include work by Zorana Ivcevic and colleagues as well as Thalia Goldstein and Ellen Winner. They have all conducted work on emotional intelligence, theory of mind, and empathy in actors and their relationship to creativity. 

Thank you for suggesting the review of the work of these relevant scholars. In the first version of the manuscript, we had endeavored to keep the literature as tightly focused on professional actors as possible, but we do also see the importance of bringing in work that focuses on the benefits of earlier acting training during childhood, adolescence, and college. In addition, we found the very recent 2020 paper from Ivcevic with visual artists interesting and relevant and included it in the review as well. We have now added a new paragraph on pp. 7-8 of the manuscript within the Summary of Extant Work section that reviews these relevant pieces, as well as another recent paper from Gentzler and colleagues that was recommended by reviewer 2 (see comment #23). This paragraph now reads: 

Another recent study specifically of student actors [20] examined the emotional attributes of undergraduate acting majors as compared to undergraduate students without acting experience. These researchers found that actors reported higher temperamental sadness and fear, but more positive viewpoints related to the experience of these negative emotions. In addition, student actors were more capable than other undergraduates at identifying facial expressions related to pride, but less capable than other undergraduates at identifying facial expression related to anger. This finding is related to another recent piece from Ivcevic and colleagues [21] who found that, despite the strong negative correlation at the population level between psychological vulnerabilities such as anxiety and depression and psychological resources such as self-acceptance and hope, creative experts (i.e., fine arts faculty) exhibited simultaneously high levels of both psychological vulnerabilities and resources, implying that creative experts may be fruitfully utilizing both their negative and positive psychological attributes to support their artistic expression. These findings are supported by a relatively long line of psychological research from scholars such as Thalia Goldstein and Ellen Winner [22] [23] [24] [25] who have shown that arts education, and specifically training in acting techniques, can support children’s development of emotional regulation, theory of mind, and other positive psychological attributes, including the capacity to safely express negative emotions. In our view, these perennial findings from the developmental and educational literature concerning the benefits of acting training for children further imply that expert of professional actors may not only benefit from such positive psychological attributes that they develop during their training but may actually require those attributes for success in their expert work.

8. Lastly, a careful edit will catch inconsistencies in citation formatting and reference formatting.

Thanks for pointing this out, and we agree that there were a number of formatting issues in the earlier version of the manuscript. In this revision, we have carefully reformatted the manuscript in order to best conform to the style of PLOS ONE. We believe all the mistakes have now been fixed. 

9. The abstract is well organized and concise. It would be helpful to include the number of actors, student actors and non-actors in the abstract as well.

We have now listed the N’s for each of the groups in the Abstract.

10. This section is generally well organized. Providing an historical overview positions the current study and highlights the importance of this study. In the discussion on empathy the authors should include research findings by Winner and Goldstein. They also examined empathy and theory of mind in actors.

Please see your comment #7 above for details about how this important line of work was addressed in the revised manuscript. 

11. This section is detailed and situates this study so that it could be easily replicated. The measures and the statistical analyses were detailed. Can the authors provide more information on the control sample? The control group also clearly engages in creative activities given their responses. 

Thank you for your positive appraisal of our Methodology section in regard to replicability and specificity. 

We agree that the non-zero amount of performing arts activities within the comparison group is important to explicitly describe in the Participants section. In this revision, we have added a paragraph on page 13 of the manuscript that reads:

Although this sample was collected as a non-acting comparison group, we did not require that these participants should have had zero history of activities within the performing arts. Indeed, some history of “little-c” [45] creative activities is likely to be expected of nearly any sample. However, as will be presented in the Results section of this paper (see Table 2 for standardized descriptive statistics), this comparison group did report statistically and practically significantly fewer creative activities than the other groups for every creative domain measured, with the greatest differences being within the performing arts domain. 

12. In combination with the tables and figure, the results were fully explained. The authors also offer solid reasons for their statistical choices. The summary results for each measure was very helpful. The overall findings were summarized succinctly. The limitations were relevant. As stated by the authors, longitudinal work is needed to understand the large attrition rate as student actors move into a professional career. 

Thank you very much for these kind words about our work.

13. It would be helpful to provide a stronger statement differentiating student actors from professional actors, especially because student actors who are trained in a conservatory setting are working in stable environments, hence their willingness to take risks would be higher. Age further influences some of these differences, a factor that was minimally addressed in this study.

Thank you for pointing out this area of the Discussion where we could have more thoroughly explained our findings. We have now added, on page 37-38 in the section on Volatility, a paragraph discussing the differences in environmental stability among professional and student actors, as well as the potential influence of age, and calling for longitudinal work related to those issues. The paragraph reads: 

In contrast to the highly volatile environment within which professional actors are situated, student actors appear much more likely to be working in stable and supportive environments (e.g., a conservatory setting), and may not be solely responsible for their own financial well-being, relying instead on parental support or student loans. This major difference in environment may explain why professional actors displayed higher levels of Volatility in their personality than did students. Another potentially influential factor in the development of Volatility may be age and natural development, but the cross-sectional methodology utilized here was unable to disentangle to effects of aging and development, on one hand, and systematic changes in environmental stability, on the other. Future longitudinal work would be needed in order to address these open questions.

Regarding the issue of Age differentiating the professional and student actors: we did think deeply about how to include this critical variable in the models. In Footnote 1, we share results of a LASSO model that included Age, which was capable to identifying professional and student actors with a 96% accuracy. The issue from our perspective was that the variance accounted for by Age washed out the influence of the psychological attributes that were the focus of this investigation. So, we decided to remove Age and other demographic variables from the LASSO model. We explain this issue, also on page 38 of the manuscript, in the following sentences:

As can seen in Footnote 1 of this paper, when age was included in the LASSO model that sorted professional from student actors, the classification rate was much stronger (i.e., 96%). Such a finding implies that age is the clearest difference between professional and student actors, but in this investigation, we chose to report and interpret LASSO models without age, because the focus of this work was on psychological, not demographic, attributes. 

Although we did decide to downplay age in our LASSO models, we agree with you that this is a crucial finding and therefore, in this revision, we also listed it in the abstract, which now includes the classification rate for professional and student actors when age is included in the model. This sentence in the Abstract now reads: 

A cross-validated Least Absolute Shrinkage and Selection Operator (LASSO) classification model was capable of identifying actors (either professional or student) from non-actors with a 92% accuracy and was able to sort professional from student actors with a 96% accuracy when age was included in the model, and a 68% accuracy with only psychological attributes included.

14. A careful edit is required to correct inconsistencies in formatting.

Thanks again for the careful read: we have now carefully edited, and we believe have caught all formatting mistakes from the earlier submission. 

15. P. 3 – change (Kogan, 2002) to the correct formatting.

This mistake has now been fixed.

16. P. 5 – Jague – should be spelled Jaque to match the reference listing

Sorry for this mis-spelling of your colleague’s name, it has been fixed now. 

17. P. 12 – change doctoral agree to doctoral degree

Fixed.

18. p. 15 – Likert is incorrectly spelled

Fixed. 

19. p. 31 – there is a double wording of “that suggests that suggests”

Fixed. Thank you again for the detailed comments. 

Reviewer 2 Comments:

20. Data Availability- The authors note in their submission information that data is available on their site, however this not mentioned in the actual manuscript.

Thanks for pointing this out. We have now added a sentence on p. 12 of the manuscript immediately under the “Methodology” heading explaining how to access the data. We also have made the text-mining models used to score the DT tasks free and easy to use on our website, so we are happy to share it widely with the field. This sentence reads:

To facilitate replicability and open science, the data collected and analyzed in this study is archived at Zenodo (DOI pending publication), and the text-mining based models used to score the divergent thinking tasks are freely available on our laboratory website (https://openscoring.du.edu/).

It should be noted that we intend to reserve a DOI for the dataset once the paper is accepted and complete a fixity and preservation-ready upload alongside publication. We are happy to provide this dataset to the field but are hoping for this paper to appear coordinated alongside, rather than after, the data is made available publicly. While we do have an associated website for this project, we believe a formal data repository is more appropriate for publication.

21. The manuscript, The Psychology of Professional and Student Actors: Creativity, Personality, and Motivation, is an interesting paper describing a newer approach to identifying group membership among actors and non-actors using a novel set of measures of creativity and other psychological dimensions. Although it has many strengths, including the LASSO approach, there are some suggestions that the authors might address to strengthen the paper.

Thank you for your generally positive appraisal of our efforts with this manuscript. Through the incorporation of your specific comments, we feel this revision is substantially improved for the readership of PLoS ONE. 

22. The “summary of extant work” section of the introduction could be streamlined. For instance, it begins as if the research is presented chronologically (e.g., “earliest work”), but then goes into negative psychological attributes, cognitive and expertise development, followed by a final paragraph that overlaps with prior content. I think taking the final paragraph in that section and incorporating it into other sections of the paper would improve flow (e.g., paragraph of Nettle’s personality and ASD components could be incorporated into the first paragraph of section or the paragraph on negative emotional attributes. 

After carefully re-reading the Summary of Extant Work section of this manuscript, we agreed that there were organizational changes to that section that were warranted. In particular, we found that the paragraph that highlighted previous work on the expertise development of actors (i.e., the work of Noice & Noice and Berry & Brown) was out of place between two paragraphs that were more related to personality differences. 

In this revision, we moved that paragraph on the expertise development literature to the end of this section, and it now is the final paragraph of the Summary of Extant Work. In our view, this organizational strategy makes sense because the expertise-related work is substantially different from most of the literature, but is highly related to our current work, given the focus on studying expert and professional actors. 

In addition, based on your comment #23 and Reviewer 1’s comments #10 and 7, we also added a paragraph to this section that brings in the work of Ivcevic, Gentzler, as well as Goldstein and Winner. Because many of these pieces generally focus on the benefits of artistic training during childhood, adolescence, and college, we found they formed a logical lead-in to the expertise work, and therefore we situated that paragraph directly before the paragraph focusing on Noice & Noice and Berry & Brown. 

23. Also, worth possibly including is the recent Gentzler, DeLong, and Smart (2019) for their comparisons of student acting majors and non-actors on a variety of psychological dimensions. For instance, they assessed temperament which can be regarded as similar to personality.

Thank you very much for pointing out this paper, which we were not aware of when writing the previous version of this manuscript. We have now specifically reviewed it on p. 7 of the manuscript in the Summary of Extant Work section. The review related to this paper reads as follows: 

Another recent study specifically of student actors [20] examined the emotional attributes of undergraduate acting majors as compared to undergraduate students without acting experience. These researchers found that actors reported higher temperamental sadness and fear, but more positive viewpoints related to the experience of negative emotions. In addition, student actors were more capable than other undergraduates at identifying facial expressions related to pride, and less capable than other undergraduates at identifying facial expression related to anger.

24. In the divergent thinking assessment section, I thought the points made by the authors as to why that measure could be a useful indicator of performance for actors was good. However, I think there is much to be elaborated. First, has that measure actually been assessed among actors? It’s lack of use for predictive power for determining group membership is made clear, but if it has been assessed as an outcome among participants is unclear as written.

Thank you for pointing out this area of the paper where our previous description could have been more thorough. As far as we are aware, DT tasks have never before been administered to professional actors (at least as part of a published research study). However, some available evidence does suggest the appropriateness of these tasks for this population. We have now inserted three detailed sentences on p. 9 of the manuscript in the Divergent Thinking Assessment section to explain this issue. These sentences read:

As far as we are aware, DT assessments have not yet been systematically administered to professional actors as part of research study, however, some initial evidence that DT measures are sensitive to acting training is available in the field. For example, Sowden and colleagues [35] demonstrated that improvisation exercises could improve the DT of elementary school students, suggesting that DT measures may be suitable for identifying individuals with acting training. In this investigation, performance assessment of DT is included, as a key way to extend past work.

25. More detail about the personality facets could be included. It’s unclear if the authors’ comments about those that load onto the Big 5 is in reference to the emotional intelligence and grit previously mentioned, or if they are referring to other facets. Given that the Big5 dimensions are often broken down to the facet-level, more clarity is warranted.

We agree that, although we had these specific delineations in the Method section in the previous version of the manuscript, a greater level of specificity was warranted earlier on. We have now inserted this specific information on pp. 10 in the manuscript, in the section headed “Richer array of self-report questionnaires”. These newly inserted sentences read: 

Finally, although the Big 5 personality attributes have previously been investigated in professional actors [13], the more finely-grained analysis of personality facets (two of which load on each of the Big 5 [39]) has never before been examined with actors. More specifically, the Big 5 dimensions of Neuroticism, Agreeableness, Conscientiousness, Extraversion, and Openness can be further delineated into 10 facets: Neuroticism contains both Volatility and Withdrawal; Agreeableness is divided into both Compassion and Politeness; Conscientiousness contains Industriousness and Orderliness; Extraversion has Enthusiasm and Assertiveness; and Openness is divided into Intellect and Openness. In this study, each of these extensions to past work are included. 

26. A clear statement about the main benefits of the project is warranted in the introduction. The first time the authors describe the benefits concretely, is in the final paragraph of the paper. The authors begin the manuscript with noting the research questions that stem from comparing the unique lifestyles of actors to non-actors, but it’s unstated as to why that matters. Although I agree it’s a great research question quite interesting to study, briefly mentioning the benefits of a study like this would greatly strengthen the selling point of the paper.

Thank you for pointing out this aspect of the presentation of our work that could improve. In this revision, we have specifically delineated the potential benefits of a study such as this one on pp 3-4 of the manuscript in a paragraph that mirrors the sentiments from the Conclusion section, but does so with a more introductory tone. This paragraph reads: 

Given this current characterization of the acting profession, with its extremely high-risk features, the question becomes pertinent: what psychological attributes distinguish those individuals who have dedicated themselves to the acting profession from those that have not? And when professional and student actors are jointly considered, what psychological attributes may differentiate those groups, and potentially contribute to professionals’ persistence and success? By closely examining and working to understand the psychological attributes that support actors in their professional work, or their development of expertise as a student, it is our intention to present findings that are not only interesting for the psychological research community but also for professional and student actors themselves. In addition, educators who seek to train student actors who may one day become professional may benefit from such an investigation, because it intends to highlight the dimensions on which professional and student actors differ or are similar, possibly informing pedagogical decisions. Finally, organizations that employ, serve, or represent actors (e.g., theater companies; talent agencies; labor unions) may find value in such an investigation, because it would delineate the strengths and further needs of actors and the acting community. 

27. Regarding the undergraduate acting majors, what year in the program were they? Do you have any indicators of how much experience those students had on stage (e.g., number of productions performed in)? I would think that those who participated in more productions would be better actors compared to less experienced students and may have important implications for the other qualities measured. 

Thank you for pointing out the important need to report the year-in-program for the students in this sample. We have now included these specific descriptive statistics on page 14 of the manuscript within the Participants section. As you will see, the sample was relatively evenly split among first, second, third, and fourth years, with a smaller number reporting taking a fifth year. 

We also agree that investigating the number of performances a student has participated in as influencing their psychological attributes may be an important avenue for research. This was the hallmark of the classic Stacey & Goldberg (1953) study as well. In these data, we found no statistically detectable effect of year-in-program on the performing arts activity measure within the undergraduate sample. In addition, you will see in Table 2 that we found that the student actors reported participating in slightly more performing arts activities than the professional actors: probably because of the increased opportunities to perform in the university space compared to the professional context. The performing arts activity measure we utilized seems to have been highly sensitive to amateur performance activities—or perhaps universities are more inclusive in the number of undergraduates who get to participate in each show then they were in 1953 when Stacey and Goldberg did their study—so we did not replicate Stacy and Goldberg’s findings related to differences in number of roles within the undergraduate sample itself.

28. Also, was there an index to confirm the non-acting participants had not performed in any productions before?

The index that we would point to as appropriate for this purpose and interpretation is the performing arts scale of the Inventory of Creative Activities. In our sampling, we did not require that those MTurk participants in the comparison group have a totally zero history of performing arts activities. In fact, we understood that a general population sample may have some history of “little-c” amateur performance over the last ten years: which was the period that the performing arts activities measure captured. However, as can be seen from Table 2, the comparison group reported far fewer experiences with the performing arts than did the student or professional actors, clearly marking them as non-actors. 

We understand that this issue should be made more explicit in the manuscript, and have now added the following text on p. 13 of the manuscript, within the Participants section:

Although this sample was collected as a non-acting comparison group, we did not require that these participants should have had zero history of activities within the performing arts. Indeed, some history of “little-c” [45] creative activities is likely to be expected of nearly any sample. However, as will be presented in the Results section of this paper (see Table 2 for standardized descriptive statistics), this comparison group did report statistically and practically significantly fewer creative activities than the other groups for every creative domain measured, with the greatest differences being within the performing arts domain. 

29. Also, it would help to know if there was any difference in the students who might be “theatre” vs “acting” students. For example, could there be students with different training/ coursework (e.g., vocal musical training for musical theatre vs. vocal training for speech alone), creating subgroups within the major? If there are differences, it would be a nice addition to know if there are any differences in the students’ trainings and experience and any implications of that. If there are no differences and “theatre and acting” is really the same program at U of Denver, it would be useful to simply adjust the wording in the paper to indicate that’s just what the program is called.

Thank you for pointing out this additional nuance within the student actor sample. All of the students within the sample had been enrolled in an acting class, and had majors declared within the area of theater, but the concentrations of these majors were in three sub-areas: acting, musical theater, or directing/playwriting. We found no statistically detectable differences in our measures across these three concentrations, and the sample size of those who did provide us with their concentration (n = 78) was too small to fully explore the patterns broken down by concentration with LASSO. 

In order to make the readership fully knowledgeable about the concentrations of the students within our sample, we added specific descriptives related to this issue on p. 14 of the manuscript within the Participants section. 

30. I thought that the description of the LASSO technique in the results section was thorough and easy to follow. I found it interesting that some variables that showed no univariate differences between groups were selected for the LASSO model predicting actor status. I think some elaboration is needed to explain why that happened, and its implications in future research for statistical methods and for understanding actors from non-actors.

We agree that the differences between the univariate results, and the results of the more nuanced LASSO models, warrant further discussion for the readers. Whenever a new technique is being introduced in an area of research it is important to explain the interpretation of the coefficients as thoroughly as possible, so thank you for showing us an area in which we could improve. 

In order to make the differences between the two as clear as possible, and to build an argument why the LASSO coefficients are likely more useful than the univariate tests, we have now inserted a fully new paragraph on pp. 29-30 of the manuscript, in the section describing and introducing the LASSO technique. This paragraph now reads:

In interpreting the results of the LASSO models in this study, and especially in interpreting how the results of the LASSO models differ from the results of the univariate tests above, it is important to consider that the LASSO coefficients are meant to represent the predictive capacity of that variable with all other variables in the model controlled for. The results of the LASSO models below can be substantively different than the univariate results above for this reason. For instance, it is possible (and even likely) that some variables that exhibited univariate differences above will be dropped from the LASSO models due to their lack of predictive power over and above the other included variables. The inverse is also possible: just because a variable did not display univariate differences above does not mean that it is not a useful predictor of acting, when combined with the other variables in the model. For these methodological reasons, we would argue that the LASSO coefficients calculated here are a more true-to-life depiction of how professional and student actors differ from non-actors, or how professional and student actors differ from each other. This is because the various psychological attributes studied here do not, in truth, operate in an isolated or univariate way. Instead, these psychological attributes operate in conjunction with one another in order to support individuals in their creative development and profession. 

31. For some findings, descriptive implications are missing from the results or discussion (e.g., emotional intelligence measures, tolerance of uncertainty). For instance, a description of implication regarding the unexpected finding at the univariate level for emotional intelligence and the more anticipated finding with LASSO is warranted. 

Thank you for pointing out that we could add more and richer implications of many of our findings within the Results and Implications section. We personally prefer specific implications being delineated directly within the Results section and are happy to include some more in specific spots—especially for the self-report measures you mentioned—within the Results section in this revision. 

For example, for the specific issue related to Emotional Intelligence that you mentioned, we added a further explanation of the finding both in the Univariate Results section and in the LASSO results section. 

On p. 25 of the manuscript, in regards to the univariate finding related to Emotional Intelligence, we added the following explanation: 

Another potential explanation of this finding is that professional actors may be better calibrated in terms of their true levels of Emotional Intelligence, whereas non-actors and students may have exhibited a self-report bias that led them to over-estimate their true levels of Emotional Intelligence, therefore moving their group means above the professional actors’ group mean. 

On p. 34 of the manuscript, in the section where the LASSO results related to Emotional Intelligence (sorting professional from student actors) are presented, we also added the following further implication:

As previously discussed in relation to the univariate findings above, the greater Emotional Intelligence of student actors than professional actors could potentially be explained by a self-report bias in students and better calibration to truth in professionals; or potentially it could relate to a need within the professional acting community to lessen emotional awareness, given their difficult and economically unstable lifestyle, in comparison to the relatively stable lifestyle of students.

32. Additionally, the implications regarding findings with tolerance of uncertainty are not elaborate. For example, the finding that students are more tolerant of uncertainty than professionals is not surprising to me given that students are less likely to be solely responsible for their own livelihood (e.g., receiving financial aid or assistance from caregivers while in college). 

We agree entirely with your synopsis and have now inserted this specific discussion on p. 24 of the manuscript immediately following the presentation of the univariate differences on the Intolerance of Uncertainty scale. The sentence we inserted reads: 

This pattern, in which student actors exhibit less Intolerance of Uncertainty than do professional actors, is perhaps not surprising given that they are situated within university contexts that are potentially more likely than the professional context to be personally supportive of individuals, and they are also much less likely than professional actors to be solely responsible for their livelihood (e.g., student actors may rely on financial aid or assistance from caregivers). 

We also inserted the additional explanation of the closely related finding from the LASSO models, on p. 34 of the manuscript: 

The Inhibitory Anxiety scale of the Intolerance of Uncertainty measure was selected for this model and weighted in the positive direction, implying that professional actors reported more inhibition in response to uncertainty than did student actors, who were very tolerant of uncertainty: a finding that appears to fit with the widely held conceptualization of student actors’ lifestyle as relatively stable and well-supported (either by caregivers or financial aid), and professional actors’ lifestyle as unstable and under-resourced.

33. Adding brief explanations for why the separate dimensions of emotional intelligence, personality, etc. were found would be very useful.

Thank you again for pushing us to add more implications to the results related to the LASSO models, which we admittedly kept brief in the previous version of the manuscript. In this revision, we have augmented the implications surrounding the LASSO results both for the model that sorted actors from non-actors and for the model that sorted professional from student actors. These augmented implications were added from pages 30-35 in the manuscript. 

As one example, we deeply increased the discussion of implications around the personality facets that were predictive of being an actor (i.e., sorting actors from non-actors), and tied these findings to some existing literature. One new paragraph that relates to these findings appears on p. 32 of the manuscript and reads: 

The personality facets of Openness, Orderliness, and Compassion were also selected for the model and weighted positively, while the Intellect personality facet was selected for the model but weighted negatively. Both the findings related to Openness and Compassion fit with general expectations reviewed here, in that actors would be expected to be generally open to new experiences, and compassionate to others, in order to develop the characters they are called upon to embody and perform. The positive weighting of the Orderliness facet of the Conscientiousness dimension was perhaps less expected, especially given the negative weighting of Industriousness. Taken together, these findings imply that actors generally prefer to follow a set schedule and prefer tidiness above disorder (positive weighting of Orderliness); while actors also struggle to carry out plans, make major decisions, and resist being distracted (negative weighting of Industriousness and Intellect). In our view, this set of findings highlights a general personality that may be able to simultaneously support the rigorous demands of a rehearsal schedule, while also allowing for the mind-wandering cognitive style that has long been associated with original thought [76].

34. A few typos are apparent (e.g., missing punctuation after citation brackets, missing “s” for the plural form of the word “responses”, spelling of “facets” as “facts”).

Thank you for pointing out these mistakes. In this revision, we have endeavored to catch all typos, and certainly fixed those you point out: “responses” on p. 16 and “facets” on page 20. In addition, all in-text citations and accompanying punctuation have been proofread.

---

## [Decision Letter · Decision Letter 1]

23 Jul 2020

PONE-D-20-02455R1

The Psychology of Professional and Student Actors: Creativity, Personality, and Motivation

PLOS ONE

Dear Dr. DUMAS,

Thank you for submitting your manuscript to PLOS ONE. After careful consideration, we feel that it has merit but does not fully meet PLOS ONE’s publication criteria as it currently stands. Therefore, we invite you to submit a revised version of the manuscript that addresses the points raised during the review process.

We look forward to receiving your revised manuscript.

Kind regards,

Paolo Roma

Academic Editor

PLOS ONE

Reviewers' comments:

Reviewer's Responses to Questions

**Comments to the Author**

1. If the authors have adequately addressed your comments raised in a previous round of review and you feel that this manuscript is now acceptable for publication, you may indicate that here to bypass the “Comments to the Author” section, enter your conflict of interest statement in the “Confidential to Editor” section, and submit your "Accept" recommendation.

Reviewer #1: All comments have been addressed

Reviewer #2: (No Response)

2. Is the manuscript technically sound, and do the data support the conclusions?

Reviewer #1: Yes

Reviewer #2: Yes

3. Has the statistical analysis been performed appropriately and rigorously? 

Reviewer #1: Yes

Reviewer #2: Yes

4. Have the authors made all data underlying the findings in their manuscript fully available?

Reviewer #1: Yes

Reviewer #2: Yes

5. Is the manuscript presented in an intelligible fashion and written in standard English?

Reviewer #1: Yes

Reviewer #2: Yes

6. Review Comments to the Author

Reviewer #1: Thank you for the changes that were requested by both reviewers and for your thoughtful and thorough revision. Your paper is very strong and very important.

Reviewer #2: The manuscript, The Psychology of Professional and Student Actors: Creativity, Personality, and Motivation, is an interesting paper describing a newer approach to identifying group membership among actors and non-actors using a novel set of measures of creativity and other psychological dimensions. The authors presented a revised manuscript that addressed prior critiques meaningfully, all of which improved the manuscript thus far. Below are a few remaining areas to consider for revision to further strengthen the paper.

1. The organization about self-report measures is much improved from the prior version. It is much clearer how personality dimensions are being advanced with this study, and how grit and other creative activities would be useful. However, it would be beneficial to have a little more description about grit and creative activities to further justify their specific inclusion in the study, beyond their novelty.

2. Thank you for providing the frequency of participants by year in program. I agree with the author’s comment about university productions potentially providing more opportunities for involvement making their number of productions similar. Were there any questions about the roles the students and professionals had in each production? Perhaps a student was in many productions, but always part of the chorus instead of a lead. Any data about the depth of actors’ roles would be another great addition to the manuscript.

3. The addition about the natural development of volatility was a good point. However, I think it opens the possibility that other developmental patterns from adolescence to adulthood may be useful in distinguishing student vs professional actors, such as with the other-emotional appraisal construct. For example, typical brain development of the neurologic system responsible for seeking rewards is highly influenced by the rewards of peers, and develops earlier than the control system responsible for self-management which continues developing into the mid-20s. Thus, the college actors might be attuned to others’ emotions and more likely to react to them than their professional counterparts who have a fully developed control system that can ignore their emotional awareness as necessary. This example and other developmental patterns could be considered for their effect on identifying actor group membership. If interested in brain development of these systems and social implications, you might check some of Steinberg’s work on social neuroscience (e.g., 2008, 2010).

Minor revision

1. Table 2 concisely offers a substantial amount of information in a small space. However, the significant group differences are not always obvious. It might be worthwhile to denote which groups were significantly different (e.g., bold the M & SDs so you don’t have to add more physical text).

7. PLOS authors have the option to publish the peer review history of their article (what does this mean?). If published, this will include your full peer review and any attached files.

Reviewer #1: **Yes: **Paula Thomson, PsyD

Reviewer #2: No

---

## [Author Response · Author response to Decision Letter 1]

29 Jul 2020

Reviewer 1 (Paula Thomson) Comments: 

1. Thank you for the changes that were requested by both reviewers and for your thoughtful and thorough revision. Your paper is very strong and very important.

Thank you very much Paula for your careful read of our manuscript and your kind words about the quality of our effort. We hope that this paper is useful to our field and can serve both communities involved in this research: social scientists and performing artists. 

Reviewer 2 Comments:

2. The manuscript, The Psychology of Professional and Student Actors: Creativity, Personality, and Motivation, is an interesting paper describing a newer approach to identifying group membership among actors and non-actors using a novel set of measures of creativity and other psychological dimensions. The authors presented a revised manuscript that addressed prior critiques meaningfully, all of which improved the manuscript thus far. Below are a few remaining areas to consider for revision to further strengthen the paper.

Thank you again for your thoughtful and detailed comments on this manuscript. We are grateful for the time and attention you have paid to our work, and believe that the revisions we have enacted in response to your comments have substantially improved this manuscript for the readership of PLoS ONE. 

3. The organization about self-report measures is much improved from the prior version. It is much clearer how personality dimensions are being advanced with this study, and how grit and other creative activities would be useful. However, it would be beneficial to have a little more description about grit and creative activities to further justify their specific inclusion in the study, beyond their novelty.

Thank you for identifying this point in the manuscript where greater detail was warranted. In this revision, we have added specific, but concise, details about the hypothesized relevance of both the Grit and Creative Activities scale within the part of the paper headed “Richer array of self-report questionnaires”, which is a sub-section of the “Promising Areas to Extend Past Work” section. These details appear on page 10 of the manuscript and read:

Although many psychologically interesting and well-validated self-report scales have been previously included in research on actors, many relevant constructs remain to be included. For example, self-reported creative activities in domains in which an individual is not a professional (e.g., creative visual arts activities for actors) have previously been shown to be predictive in creativity research [36]. For instance, it may be reasonable to hypothesize that actors, given the creative nature of their work, will engage in more creative activities than non-actors even in domains (e.g., literature; music) that are not directly within the area of acting. Relatedly, it could also be, that because actors’ work demands creative thinking, they may tend to avoid expending creative effort in other more quotidian domains such as cooking.

In addition, motivational constructs such as Grit [37] that have come to greater attention in the literature recently, have never been examined with professional actors before. In the context of the acting profession, where financial security can be lacking, and rejection (i.e., not booking an auditioned-for role) is commonplace, motivational attributes such as perseverance in the face of adversity and consistency of interest in one’s chosen profession—two principal facets of Grit [37]—appear likely to be relevant.

4. Thank you for providing the frequency of participants by year in program. I agree with the author’s comment about university productions potentially providing more opportunities for involvement making their number of productions similar. Were there any questions about the roles the students and professionals had in each production? Perhaps a student was in many productions, but always part of the chorus instead of a lead. Any data about the depth of actors’ roles would be another great addition to the manuscript.

This is an excellent point and thank you for raising it. Unfortunately, we did not collect any data concerning the centrality or high-profile nature of the roles the student or professional actors were cast in. So, in this data set we do not know which of roles each actor (professional or student) booked were lead roles and which were ensemble roles. We are currently undertaking a follow-up study where we are investigating the type of role (e.g., comedic/dramatic; stage/screen) actors tend to book, and the psychological patterns that predict being cast in those differing types of roles. However, we did not think about asking about leading roles until you raised this point, so thanks!

In order to adequately address this point in this revision, we added two detailed sentences about this issue within the Limitations section of the manuscript. These sentences appear on page 40 of the paper and read: 

Finally, it is possible that student or professional actors that tend to book higher- or lower-profile roles (e.g., lead roles versus ensemble roles) also differ systematically in their psychological attributes. In this study, we did not ascertain among the student or professional actors which individuals in the sample were being cast in more central roles, and this appears to be a fertile ground for future research that drills deeper into the psychology of student and professional actors.

5. The addition about the natural development of volatility was a good point. However, I think it opens the possibility that other developmental patterns from adolescence to adulthood may be useful in distinguishing student vs professional actors, such as with the other-emotional appraisal construct. For example, typical brain development of the neurologic system responsible for seeking rewards is highly influenced by the rewards of peers, and develops earlier than the control system responsible for self-management which continues developing into the mid-20s. Thus, the college actors might be attuned to others’ emotions and more likely to react to them than their professional counterparts who have a fully developed control system that can ignore their emotional awareness as necessary. This example and other developmental patterns could be considered for their effect on identifying actor group membership. If interested in brain development of these systems and social implications, you might check some of Steinberg’s work on social neuroscience (e.g., 2008, 2010).

Thank you for raising this interesting point, which we would not have thought to include without your prompting. Because social neuroscience is not a main area of expertise for any of us, you will see that we adapted your wording of the issue, and the two Steinberg citations you recommend, to fit within the paper. In the manuscript, the most in-depth description of the Other-emotional Appraisal finding (which was found in the LASSO model sorting professional from student actors) was on pages 35-36, in the final paragraph of the Results section. We added a few detailed sentences on this page, following the description you recommended. These sentences read: 

Evidence from social neuroscience research may also relate to this finding of greater Other-emotional Appraisal in student actors. It is understood within the social neuroscience literature that the neurological system responsible for seeking rewards is highly influenced by the social rewards available from peers, and that this system develops relatively early in adolescence; in contrast, the neurological control system responsible for self-management is likely not fully developed until an individual’s mid-20s [87] [88]. Therefore, it may be that undergraduate actors are closely emotionally attuned to their peers in their theater productions given their as-yet-not-fully-developed neurological system, while professional actors have the adult neurological capacity to control or ignore their emotional awareness when it is necessary.

6. Table 2 concisely offers a substantial amount of information in a small space. However, the significant group differences are not always obvious. It might be worthwhile to denote which groups were significantly different (e.g., bold the M & SDs so you don’t have to add more physical text).

Thank you for this pragmatic suggestion to improve the presentation of descriptive statistics and mean comparisons in Table 2. Given that nearly all of the mean comparisons presented in Table 2 exhibited significance, we chose to bold the name of the variables that showed significance, rather than bolding all of the descriptives. We also added new information about this bolding scheme in the note for Table 2. This new language reads: 

Bolded variable names indicate that variable exhibited significant differences across groups at the p < .05 level.

---

## [Decision Letter · Decision Letter 2]

2 Oct 2020

The Psychology of Professional and Student Actors: Creativity, Personality, and Motivation

PONE-D-20-02455R2

Dear Dr. Dumas,

We’re pleased to inform you that your manuscript has been judged scientifically suitable for publication and will be formally accepted for publication once it meets all outstanding technical requirements.

Kind regards,

Paolo Roma

Academic Editor

PLOS ONE

Reviewers' comments:

Reviewer's Responses to Questions

**Comments to the Author**

1. If the authors have adequately addressed your comments raised in a previous round of review and you feel that this manuscript is now acceptable for publication, you may indicate that here to bypass the “Comments to the Author” section, enter your conflict of interest statement in the “Confidential to Editor” section, and submit your "Accept" recommendation.

Reviewer #2: All comments have been addressed

2. Is the manuscript technically sound, and do the data support the conclusions?

Reviewer #2: Yes

3. Has the statistical analysis been performed appropriately and rigorously? 

Reviewer #2: Yes

4. Have the authors made all data underlying the findings in their manuscript fully available?

Reviewer #2: Yes

5. Is the manuscript presented in an intelligible fashion and written in standard English?

Reviewer #2: Yes

6. Review Comments to the Author

Reviewer #2: The manuscript, The Psychology of Professional and Student Actors: Creativity, Personality, and Motivation, is a great manuscript describing actors and non-actors with a unique set of measures. The authors presented a very strong revised manuscript that addressed prior critiques meaningfully and improved the manuscript. Thank you for your thorough revision. Your manuscript substantially adds to the literature.

7. PLOS authors have the option to publish the peer review history of their article (what does this mean?). If published, this will include your full peer review and any attached files.

Reviewer #2: No

---

## [Editor Report · Acceptance letter]

12 Oct 2020

PONE-D-20-02455R2 

The Psychology of Professional and Student Actors: Creativity, Personality, and Motivation 

Dear Dr. Dumas:

I'm pleased to inform you that your manuscript has been deemed suitable for publication in PLOS ONE. Congratulations! Your manuscript is now with our production department. 

Kind regards, 

on behalf of

Prof. Paolo Roma 

Academic Editor

PLOS ONE